# Synthesis and evaluation of historical meridional heat transport from midlatitudes towards the Arctic

Yang Liu[1,2], Jisk Attema[1], Ben Moat[3], and Wilco Hazeleger[1,2]

[1]Netherlands eScience Center, 1098 XG, Amsterdam, The Netherlands
[2]Wageningen University, 6708 PB, Wageningen, The Netherlands
[3]National Oceanography Center, SO14 3ZH, Southampton, United Kingdom

**Correspondence:** Yang Liu (y.liu@esciencecenter.nl)

**Abstract.** Meridional Energy Transport (MET), both in the atmosphere (AMET) and ocean (OMET), has significant impact on the climate in the Arctic. In this study, we quantify AMET and OMET at subpolar latitudes from six reanalyses datasets. We investigate the differences between the datasets and we check the coherence between MET and the Arctic climate variability at interannual time scales. The results indicate that, although the mean transport in all datasets agree well, the spatial distribution and temporal variations of AMET and OMET differ substantially among the reanalyses data sets. For the ocean, only after 2010 the low frequency signals for all reanalyses products agree well. A further comparison with observed heat transports at 26.5°N and the subpolar Atlantic, and a high resolution ocean model hindcast confirm that the OMET estimated from reanalyses are consistent with independent observations. For the atmosphere, the variations among reanalyses data sets are large. This can be attributed to differences in temperature transport and geopotential energy transport. A further analysis of linkages between the Arctic climate variability and AMET shows that atmospheric reanalyses differ substantially from each other. Among all the chosen atmospheric products, ERA-Interim and JRA55 results are most consistent with results obtained with coupled climate models. For the ocean, ORAS4 and SODA3 agree well on the relation between OMET and sea ice concentration (SIC), while GLORYS2V3 deviates from those data sets. The regressions of multiple fields in the Arctic on both AMET and OMET suggest that the Arctic climate is sensitive to changes of meridional energy transport at subpolar latitudes in winter. Our study suggests, since the reanalyses products are not designed for the quantification of energy transport, the AMET and OMET estimated from reanalyses should be used with great care, especially when studying variability and interactions between the Arctic and midlatitudes beyond interannual time scales.

## 1 Introduction

Poleward meridional energy transport, both in the atmosphere (AMET) and ocean (OMET), is one of the most fundamental aspects of the climate system. It is closely linked to changes of weather and climate at different latitudes. The quantification of AMET and OMET has been studied extensively. Dating back to 1970s, many efforts were made to reproduce the AMET and OMET with very limited observational data available (Vonder Haar and Oort, 1973; Oort and Vonder Haar, 1976). After entering the satellite era, further progress has been made during the recent two decades. Using the radiation at top of the atmosphere and the reanalyses data, a complete picture of AMET and OMET is given by Trenberth and Caron (2001). Following their

work, rapid progress was made using similar methodologies and new data sets of observations (Ganachaud and Wunsch, 2000, 2003; Wunsch, 2005; Fasullo and Trenberth, 2008; Zheng and Giese, 2009; Mayer and Haimberger, 2012). Nevertheless, these estimations still suffered from problems like mass imbalance, unrealistic moisture budget, coarse resolution, and sparseness of observations (Trenberth, 1991; Trenberth and Solomon, 1994). Fortunately, recent improvements in numerical weather and

ocean models and increased data coverage of observations provide a basis to improve estimates of AMET and OMET. There is an increase of available reanalyses products, increase in resolution, length of the time span that is covered and increase of components of the Earth system that are included in the products (Dee et al., 2011; Gelaro et al., 2017; Harada et al., 2016; Balmaseda et al., 2013; Ferry et al., 2012b; Carton et al., 2018). It is very promising to have better quantification of AMET and OMET using the latest reanalyses data sets.

To support our elaboration on MET, we also study AMET and OMET in relation to climate variability at interannual time scales in the Arctic region. In recent decades, the Arctic is warming twice as fast as the global average (Comiso and Hall, 2014; Francis et al., 2017). This phenomenon is known as Arctic Amplification (AA) and it has an impact far beyond the Arctic (Miller et al., 2010; Serreze and Barry, 2011). In order to understand the warming, the process behind the AA, its wider consequences and to make reliable predictions of the Arctic climate, it is crucial to understand the Arctic climate variability. Among all the

factors responsible for the variability in the processes described above, meridional energy transport (MET), from midlatitudes toward the Arctic, plays a significant role (Graversen et al., 2008; Kapsch et al., 2013; Zhang, 2015). There is a large volume of published studies describing the impact of AMET and OMET on the variation of sea ice and the warming in the Arctic. Using reanalysis data, Yang et al. (2010) show that the poleward AMET is linked with the evolution of temperature in the free troposphere at decadal time scales. By separating the planetary and synoptic-scale waves, Graversen and Burtu (2016) show

that the latent heat transport, as a component of AMET, influences the Arctic warming with reanalysis data. Gimeno-Sotelo et al. (2019) studied the moisture transport for precipitation with reanalysis data and observation data, and show the moisture sources for the Arctic region is linked with interannual fluctuations in the extent of Arctic sea ice. Nummelin et al. (2017) analyse the linkages between OMET, Ocean Heat Content (OHC) and AA through the simulations within the Coupled Model Intercomparison Project Phase 5 (CMIP5). They report enhancement of OMET as a result of heat loss in the subpolar ocean and

the contribution of OMET to the AA through the increasing of OHC in the Arctic ocean. Also by analyzing CMIP5 simulations, Sandø et al. (2014) show a large impact of heat transport in the Barents Sea on sea ice loss. However, ocean reanalyses don't show a clear sign of AA in Arctic OHC increases (Mayer et al., 2016; Von Schuckmann et al., 2016). Consequently, increasing knowledge on poleward AMET and OMET at subpolar and polar latitudes will aid in understanding of AA.

Global climate models indicate a compensation between variations in atmospheric and oceanic heat transports at subpolar

and midlatitudes (Outten et al., 2018). This is indicative of positive feedbacks between the ocean and atmosphere and it has been associated with variations in sea ice by several studies (Van der Swaluw et al., 2007; Jungclaus and Koenigk, 2010; van der Linden et al., 2016). These studies all point out the connection between energy transport and the variations of the Arctic climate. However, these results are mostly based on numerical model simulations and they tend to differ among the models. In contrast to numerical modeling studies, here we intend to study AMET, OMET variability and the relation with the

Arctic in best estimates of the historical variability.

In this paper, we quantify AMET and OMET using multiple state-of-the-art reanalyses products. These are representations of the historical state of the atmosphere and ocean optimally combining available observations and numerical simulations using data assimilation techniques. Emphasis is placed on the variation of AMET and OMET from midlatitudes to the Arctic at inter-annual time scales. In contrast with earlier studies, we will compare the different reanalyses data sets. Independent observations

in the Atlantic from the Rapid Climate Change-Meridional Overturning Circulation and Heatflux Array (RAPID ARRAY) and the Overturning in the Subpolar North Atlantic Program (OSNAP) are included in the comparison. The RAPID ARRAY is a trans-basin observing array along 26.5° N in the Atlantic (Johns et al., 2011; McCarthy et al., 2015). It operates since 2004 and provides the volume and heat transport in the Atlantic basin. OSNAP is an ocean observation program designed to provide a continuous record of the trans-basin fluxes of heat, mass and freshwater in the subpolar North Atlantic (Susan Lozier et al.,

2017; Lozier et al., 2019). Moreover, a state-of-the-art NEMO-LIM2 1/12° ocean circulation / sea ice model simulation forced by Drakkar Surface Forcing data set version 5.2 (Moat et al., 2016) is also included in the comparison. It will be referred as Oceanic General Circulation Model (OGCM) hindcast in this paper. Based on the intercomparison of reanalysis data, especially with the independent observation data, we will be able to identify sources of uncertainty. To support our comparison of AMET and OMET, we also investigate the interactions between oceanic and atmospheric variations and remote responses.

The correlations between the variability of AMET and OMET and the changes in the Arctic climate are compared to literature. This is motivated by previous studies with only numerical models or a single reanalysis dataset to explain those connections (Graversen, 2006; Van der Swaluw et al., 2007; Graversen et al., 2008; Jungclaus and Koenigk, 2010; Kapsch et al., 2013).

The paper is organized as follows: Section 2 presents the data and our methodology. Results and analysis are given in Section 3. It includes AMET and OMET calculated from reanalysis data and an intercomparison of them. The correlation between the

variability of AMET and OMET, and the Arctic climate is elaborated upon in detail. Finally, remarks constitute Section 4 and conclusions are provided in Section 5.

## 2   Data and Methodology

The reanalyses data sets used in this study are introduced in this section. Moreover, the methodology for the quantification of AMET and OMET are also included in this section. The statistical tests performed in this study are elucidated in detail.

### 2.1   Reanalyses

In order to make use of observations and advanced numerical models, six state-of-the-art reanalyses data sets are used in this study. The chosen reanalyses products have high temporal and spatial resolution due to the need for the computation of energy transport (see section 2.3). It is preferable that they incorporate the latest numerical models and data assimilation schemes. As a result, we chose three atmosphere reanalyses data sets: ERA-Interim, MERRA2, and JRA55 (references below) and three ocean

reanalyses data sets: ORAS4, GLORYS2V3, and SODA3 (references below). To avoid interpolation errors and imbalances in the mass budget introduced by regridding, the calculations are based on the data from the original model grid. Note that the latest atmospheric reanalysis ERA5 from ECMWF is not included here since the model level data has not been opened to the

public yet (ECMWF, 2017). In addition, the computation is too expensive to achieve a longer time series for the study of the interannual variability of AMET using ERA5. As a synthesis, Table 1 shows the basic specifications of the reanalyses products contained in this study.

### 2.1.1 ERA-Interim

ERA-Interim is a global reanalysis dataset produced by the European Center for Medium Range Weather Forecasts (ECMWF) (Dee et al., 2011), which covers the data-rich period since 1979. It employs the cycle 31r2 of ECMWF's Integrated Forecast System (IFS) and generates data using 4D-Var assimilation with a T255 (~79km) horizontal resolution on 60 vertical levels (Berrisford et al., 2009). Compared with its preceding reanalyses, ERA-40 (Uppala et al., 2005), ERA-Interim is superior in quality in terms of the atmospheric properties like mass, moisture and energy (Berrisford et al., 2011). The improvement in observations and the ability of 4D-Var contributes a lot to the quality of the divergent wind (Berrisford et al., 2011), which is significant for the mass budget and hence the energy budget. We use the data on the original model grid, with a 0.75° x 0.75° horizontal resolution and 60 vertical hybrid model levels. We take 6-hourly data with a range from 1979 to 2016.

### 2.1.2 MERRA2

The Modern-Era Retrospective Analysis for Research and Applications version 2 (Gelaro et al., 2017), in short MERRA2, is the successor of MERRA from the Global Modeling and Assimilation Office (GMAO) of the National Aeronautics and Space Administration (NASA). It assimilates observational data with the Goddard Earth Observing System (GEOS) model and analysis scheme (Molod et al., 2015; Gelaro et al., 2017). The data is produced by a 3D-Var assimilation scheme and has a coverage from 1980 till present. Unlike most of the reanalyses products, the GEOS atmospheric model includes a finite-volume dynamical core which uses a cube-sphere horizontal-discretization (Gelaro et al., 2017). The model grid has a resolution of 0.5° x 0.625° with 72 hybrid levels. For this study, we use the 3-hourly assimilation data on the native model grid from 1980 to 2016.

### 2.1.3 JRA55

Extending back to 1958, Japanese 55-year reanalyses (JRA55) is the second reanalyses product made by Japan Meteorological Agency (JMA) (Kobayashi et al., 2015; Harada et al., 2016). JRA55 applies 4D-Var assimilation and it is generated on TL319 horizontal resolution with 60 hybrid levels. Before entering the satellite era in 1979, the assimilated upper air observations mainly come from radiosonde data. In this project we take 6-hourly data from 1979 to 2015 on the original model level, which has a horizontal resolution of 0.5625° x 0.5625° with 60 hybrid model levels.

### 2.1.4 ORAS4

Serving as the historical reconstruction of the ocean's climate, the Ocean reanalyses System 4, in short ORAS4, is the replacement of the old reanalyses system ORAS3 used by the ECMWF (Balmaseda et al., 2013). It implements Nucleus for European

Modelling of the Ocean (NEMO) as ocean model (Madec, 2008; Ferry et al., 2012a) and uses NEMOVAR as the data assimilation system (Mogensen et al., 2012). The model is forced by atmosphere-derived daily surface fluxes, from ERA-40 from 1957 to 1989 and ERA-Interim from 1989 onwards. ORAS4 produces analyses with a 3D-Var FGAT assimilation scheme and spans from 1958 to present. ORAS4 runs on the ORCA1 grid, which is associated with a horizontal resolution of $1°$ in the extratropics and a refined meridional resolution up to $0.3°$ in the tropics. It has 42 vertical levels, 18 of which are located at upper 200m. Here we skip the first two decades and use the monthly data from 1979 to 2014 to avoid the uncertainties reported by Balmaseda et al. (2013). We will use the monthly mean fields on the native model grid.

### 2.1.5 GLORYS2V3

GLORYS2V3, short for GLobal Ocean reanalyses and Simulations version 3, is a global ocean and sea-ice eddy permitting reanalyses yielded from the collaboration between the Mercator Ocean, the Drakkar consortium and Coriolis Data center (Ferry et al., 2010, 2012b). It spans the altimeter and Argo eras, from 1993 till present. The NEMO ocean model is implemented on the ORCA025 grid (approximate $0.25°$ x $0.25°$ with 75 vertical levels). The model is forced by a combination of ERA-Interim fluxes (e.g. shortwave radiation) and turbulent fluxes obtained with bulk formulae using ERA-Interim near-surface parameters. The data is generated by a 3D-Var assimilation scheme with temperature and salinity profiles assimilated from the CORA3.3 database (Ferry et al., 2012b). In this study, monthly data from 1993 to 2014 on the original ORCA025 grid is used.

### 2.1.6 SODA3

SODA3 is the latest version of Simple Ocean Data Assimilation (SODA) ocean reanalyses conducted mainly at the University of Maryland (Carton et al., 2018). SODA3 is built on the Modular Ocean Model v5 (MOM5) ocean component of the Geophysical Fluid Dynamics Laboratory CM2.5 coupled model (Delworth et al., 2012) with a grid configuration of approximately $0.25°$ x $0.25°$ x 50 levels resolution (Carton et al., 2018). To be consistent with the other two reanalyses data sets assessed in this study, the SODA 3.4.1 is chosen since it applies surface forcing from ERA-Interim. For this specific version, the 5-daily data is available from 1980 to 2015. Reanalysis data from this period on original MOM5 grid is used in this case.

### 2.2 Oceanic Observations and OGCM Hindcast

For the purpose of independent examination of the OMET calculated from reanalyses, observations of the meridional transport of mass and heat throughout the Atlantic basin are used here. We use data from the RAPID-MOCHA-WBTS program (Johns et al., 2011; McCarthy et al., 2015) and the OSNAP program (Susan Lozier et al., 2017; Lozier et al., 2019). The RAPID-MOCHA-WBTS program, which is known as RAPID ARRAY, employs a transbasin observing array along $26.5°N$ and it is in operation since 2004. The OMET from the RAPID ARRAY available to this study is from April 2004 to March 2016. The OSNAP program has an observing system that comprises of an integrated coast-to-coast array extending from the southeastern Labrador shelf to the southwestern tip of Greenland, and from the southeastern tip of Greenland to the Scottish shelf. So far, it provides OMET data from the full installation of the array in 2014 until the first complete data recovery in 2016, 21 months in

total. Although it is short to provide a good estimate of interannual variability of OMET, we include it anyway as it is a unique observation system for OMET in the subpolar Atlantic.

Apart from the RAPID ARRAY and OSNAP observational data, a high resolution hindcast of the NEMO ORCA ocean circulation model is also included here to provide more insights to the analysis since two of the chosen reanalyses products are also built on NEMO model (Moat et al., 2016; Marzocchi et al., 2015). This forced model simulation implements the NEMO ORCA global ocean circulation model version 3.6 (Madec, 2008). It is configured with ORCA0083 grid, which has a nominal resolution of $1/12°$, on 75 vertical levels. The climatological initial conditions for temperature and salinity were taken in January from PHC2.1 at high latitudes (Steele et al., 2001), MEDATLAS in the Mediterranean (Jourdan et al., 1998), and the rest from Levitus et al. (1998). It is forced by the surface fields coming from the Drakkar project,which supplies surface air temperature, winds, humidity, surface radiative heat fluxes and precipitation, and a formulation that parameterizes the turbulent surface heat fluxes and is provided for the period 1958 to 2012 (dataset version 5.2) (Brodeau et al., 2010; Dussin et al., 2016). More information about this hindcast is given by Moat et al. (2016). We take monthly mean data from the hindcast, which spans from 1979 to 2012.

## 2.3 Computation of Meridional Energy Transport

The methods for quantification of AMET and OMET with atmospheric and oceanic reanalyses are included in this section, respectively.

### 2.3.1 Energy Budget in the Atmosphere

The total energy per unit mass of air has four major components: internal energy ($I$), latent heat ($H$), geopotential energy ($\phi$) and kinetic energy ($k$). They are defined as:

$$
\begin{aligned}
I &= c_p T \\
H &= L_v q \\
\Phi &= gz \\
k &= \frac{1}{2}\mathbf{v} \cdot \mathbf{v}
\end{aligned}
\tag{1}
$$

with $c_p$ the specific heat capacity of dry air at a constant pressure ($J/(kgK)$), $T$ the absolute temperature ($K$), $L_v$ the specific heat of condensation ($J/kg$), $q$ the specific humidity $kg/kg$, $g$ the gravitational acceleration ($kg/(ms^2)$), $z$ the altitude ($m$) and $v$ the zonal/meridional wind velocity ($m/s$). The northward propagation is positive. In addition, these four quantities can be divided into three groups: the dry static energy $I + \phi$, the moist static energy $H$ and the kinetic energy $k$. A constant value of $c_p = 1004.64 J/(kgK)$ and $L_v = 2264.67 KJ/kg$ were used to compute the AMET with all the atmosphere reanalyses data sets. In addition, recently there are some improved formulations of energy budget equations proposed by Mayer et al. (2017) and Trenberth and Fasullo (2018). We use an updated formulation of AMET as a combination of the divergence of dry-air

enthalpy, latent heat, geopotential and kinetic energy transports, which is suggested by Mayer et al. (2017). Note that in this case the enthalpy transports associated with vapor fluxes are neglected.

In pressure coordinates, the total energy transport at a given latitude $\Phi_i$ can be expressed as (Mayer et al., 2017):

$$E = \oint_{\Phi=\Phi_i} \int_{p_s}^{p_t} [(1-q)c_pT + L_vq + gz + \frac{1}{2}\mathbf{v}\cdot\mathbf{v}]v\frac{dp}{g}dx \tag{2}$$

with $p_t$ the pressure level at top of the atmosphere ($Pa$) and $p_s$ the pressure at the surface ($Pa$). Since we work on the native hybrid model coordinate with each atmosphere reanalyses product, the equation can be adjusted as follows (see Graversen (2006)):

$$E = \oint_{\Phi=\Phi_i} \frac{1}{g} \int_0^1 [(1-q)c_pT + L_vq + gz + \frac{1}{2}\mathbf{v}\cdot\mathbf{v}]v\frac{\partial p}{\partial \eta}d\eta dx \tag{3}$$

where $\eta$ indicates the number of the hybrid level. Note that difference in horizontal advection schemes can influence the results.
All the chosen atmospheric reanalyses use Semi-Lagrangian advection schemes but this is not the case for MERRA2.

Unfortunately, a direct estimation of AMET based on the equations above cannot provide a meaningful energy transport obtained from reanalysis data. It has been widely reported that reanalyses products suffer from mass inconsistency (Trenberth, 1991; Trenberth et al., 2002; Graversen, 2006; Graversen et al., 2007; Chiodo and Haimberger, 2010; Berrisford et al., 2011). Spurious sinks and sources mainly come from low spatial and temporal resolution, interpolation and regridding, and data as-
similation. The interpolation from original model level to pressure level can introduce considerable error to the mass budget (Trenberth et al., 2002). Therefore we prevent interpolations onto the pressure levels and use data on the native model levels with a high temporal resolution. Trenberth (1991) provided a method to correct the mass budget through the use of the continuity equation. The method assumes that the mass imbalance mainly comes from the divergent wind fields and corrects the overall mass budget by adjusting the barotropic wind. The conservation of mass for a unit column of air can be represented as:

$$\frac{\partial p_s}{\partial t} + \nabla \int_{p_s}^{p_t} \mathbf{v}dp = g(E - P) \tag{4}$$

Where $E$ stands for evaporation and $P$ denotes precipitation. It has been noticed that big uncertainties reside in the evaporation and precipitation of global reanalyses (Graversen, 2006). Hence we use the moisture budget to derive the net moisture change in the air column, according to:

$$E - P = \frac{\partial}{\partial t}(\int_{p_s}^{p_t} q\frac{dp}{g}) + \nabla \int_{p_s}^{p_t} (\mathbf{v}\cdot q)\frac{dp}{g} \tag{5}$$

After determining the mass budget imbalance, we correct the barotropic wind fields $(u_c, v_c)$, with $u_c$ and $v_c$ indicating the correction terms for zonal and meridional wind components as a result of barotropic mass budget correction, and then calculate AMET (Trenberth, 1991). Note that all the computations regarding baratropic mass budget correction were performed in the spectral domain via spherical harmonics. Figure 1 shows the mean AMET and each component in each month at 60°N estimated from ERA-Interim.

### 2.3.2 Energy Budget in the Ocean

Unlike the atmosphere, energy transport in the ocean can be well represented by the internal energy itself. Consequently, the total energy transport in the ocean at a given latitude $\phi_i$ can be expressed in terms of the temperature transport (Hall and Bryden, 1982):

$$E = \oint_{\Phi=\Phi_i} \int_{z_b}^{z_0} \rho_0 c_{p_0} \theta \cdot v \, dz \, d\phi \tag{6}$$

where $\rho_0$ is the sea water density $(kg/m^3)$, $c_{p_0}$ is the specific heat capacity of sea water $(J/(kg°C))$, $\theta$ is the potential temperature $(°C)$, $v$ is the meridional current velocity $(m/s)$, $z_0$ and $z_b$ are sea surface and the depth till the bottom (m), respectively. A constant value of $c_{p_0} = 3987 J/(kg°C)$ was used in all the calculations of OMET. Ocean heat content (OHC, with unit $J$) is another variable that plays a role in the ocean heat budget. The total OHC between certain latitudes can be calculated by:

$$OHC = \int_{\Phi_i}^{\Phi_0} \int_{z_b}^{z_0} \rho_0 c_{p_0} \theta \, dz \, d\phi \tag{7}$$

Our computation of OMET suffers from a small mass imbalance (e.g. mass imbalance coming from the residual between precipitation and evaporation (Mayer et al., 2017)). In the ocean, with its strong boundary circulations even the smallest imbalance can lead to large errors in the heat flux. However, the barotropic correction method adopted by the atmosphere is not feasible here, as a consequence of a varying sea surface height. In oceanographic literature it is common to use a reference temperature when calculating OMET in both observations and model diagnostics (Bryan, 1962; Hall and Bryden, 1982; Zheng and Giese, 2009). Here, we also take a reference temperature $\theta_r$ $(C)$. Note that the influence from taking a reference temperature on a zonally integrated transport is smaller than that on a single strait (Schauer and Beszczynska-Möller, 2009). Then the quantification of OMET becomes:

$$E = \oint_{\Phi=\Phi_i} \int_{z_b}^{z_0} \rho_0 c_{p_0} (\theta - \theta_r) \cdot v \, dz \, d\phi \tag{8}$$

Here, we take $\theta_r$ equal to 0. Finally, operations in the "zonal" direction are different from their conventional meaning. As the three ocean reanalyses products used here are all built on a curvilinear grid, the zonal direction on the native model grid is curvilinear as well. Similar to the considerations made in Section 2.1, regridding from the native curvilinear grid to a uniform geographical grid will introduce large errors. So, we work on the original multi-pole grid and follow the native zonal directions when performing numerical operations. After applying this method the resulting OMET values are comparable to those in earlier publications (Trenberth and Caron, 2001; Wunsch, 2005; Trenberth and Fasullo, 2008). Note that since we only have access to sub-monthly data for SODA3, the computation of OMET using monthly data in ORAS4 and GLORYS2V3 could miss part of the heat transport by eddies.

## 2.4 Statistical Analysis

In order to understand the connection between MET and changes in the Arctic and compare to the results from numerical climate models or single reanalysis dataset (Graversen, 2006; Van der Swaluw et al., 2007; Graversen et al., 2008; Jungclaus and Koenigk, 2010; Kapsch et al., 2013), in the following section we performed linear regressions on multiple fields with AMET and OMET. To test the significance of the regressions, we simply use student's t-test. We decorrelate the monthly mean OMET anomalies after the implementation of low pass filter. This means the relevant significance tests are performed with time series after removing the autocorrelation.

Note that all the reanalyses data sets included in this study have short time series at monthly time scales (no more than 456 months, see Table 1). Therefore the analysis based on these data sets is not statistically significant compared with those using the output data from numerical simulations with a large time span, since the relatively short records of reanalyses do not have many samples at interannual time scales. Nevertheless, the reanalyses products are better representations of the real world. So the statistical analysis with reanalysis data is still useful to answer the questions about connections in climate system.

## 3 Results

Unless specifically noted, the results shown in this section are all based on monthly mean fields with low pass filter from 1 to 5 years.

### 3.1 Overview of AMET and OMET

Globally, MET is driven by the unequal distribution of net solar radiation and thermal radiation. The atmosphere and oceans transport energy from regions receiving more radiation to the regions receiving less. Figure 2 gives the mean of AMET and OMET over the entire time series of every product at each latitude in the Northern Hemisphere. For the atmosphere, all three datasets agree very well. The results differ a bit in amplitude but capture similar variations along each latitude. The peak of AMET is around $41°$N, after which it starts to decrease towards the north pole. In ERA-Interim and JRA55 AMET peaks at 4.45 PW at $41°$N, while in MERRA2 AMET peaks at 4.5 PW at $41.5°$N. These findings are consistent with previous work (e.g. Trenberth and Caron (2001); Fasullo and Trenberth (2008); Mayer and Haimberger (2012) and many others).

Apart from the climatology of MET, we are particularly interested in the variations across different time scales from midlatitudes towards the Arctic. The time series of AMET, integrated zonally over $60°$N, are shown in Figure 3a. The seasonal cycle is dominant in each component as expected and the phase is very similar, but differences in the amplitudes are noted. The mean AMET provided by the chosen three atmospheric reanalyses agrees well. However, their variations differ from each other. In ERA-Interim, the standard deviation (std) of AMET is 0.92 PW, while MERRA2 has a relatively large std of 0.97 PW and in JRA55 the std is 0.91 PW. Hence it can be concluded that the seasonal cycles of AMET presented by the chosen atmospheric reanalyses are similar. After removing the seasonal cycle and applying a low pass filter, neither the amplitude nor the trend of the signals agree between the data sets (see Figure 3b). The std of the AMET anomaly in ERA-Interim is 0.02 PW, while in MERRA2 it is 0.04 PW and in JRA55 it is 0.03 PW. This implies that the variation of AMET anomalies are different in the chosen data sets. We further assess the sources of the difference in the next section.

For the ocean, all the reanalyses data sets agree well at almost all the latitudes except for the OMET between $30°$N and $40°$N, where the Gulf Stream resides. The difference can be explained by the models. GLORYS2V3 and SODA3 both have been generated with eddy-permitting models while ORAS4 has not. In ORAS4, an eddy parameterization scheme from Gent and Mcwilliams (1990) is implemented. The implementation of this eddy parameterization scheme can lead to a big difference in volume transport and heat transport, compared to eddy-permitted models (Stepanov and Haines, 2014). However, in this case the computation of OMET with ORAS4 does not include the contribution from eddy-induced velocity as the fields related to the use of eddy advection schemes were not saved. The eddy-permitting reanalyses with higher resolution, like GLORYS2V3 and SODA3, are capable of addressing the large scale turbulence. It has been shown that their eddy-permitting capacity can account for the large scale eddy variability and represent the eddy energy associated with both the Gulf Stream and the Kuroshio pathways well (Masina et al., 2017). Consequently, at the latitude of the Gulf Stream (between $30°$N and $40°$N), a higher spatial variability, which represents more realistic patterns of the large scale eddy variability, is apparent in all datasets but ORAS4.

Similarly, we show the zonal integral of the OMET at $60°$N in Figure 4. Differences in amplitude and trends can be observed in the unfiltered time series. The mean OMET and the std of all the OMET time series are similar (see Figure 4a). The mean OMET in ORAS4 is 0.47 PW, in GLORYS2V3 is 0.44 PW and in SODA3 is 0.46 PW. The OGCM hindcast gives a similar mean OMET of 0.47 PW. For the std of OMET, ORAS4 and the OGCM hindcast give 0.06 PW, while GLORYS2V3 and SODA3 give 0.07 PW. In terms of the difference in the OMET time series between the chosen products, it is not surprising that large differences appear after we take a running mean of 5 years when computing the OMET anomalies. However, the large variation of OMET anomalies in Figure 4b is not noticeable from their std. Given the time series of all the chosen reanalyses, ORAS4 resembles SODA3, especially after 1998. Whereas, GLORYS2V3 is clearly different from ORAS4 and SODA3 from 1998 to 2006. The differences can be tracked in the time series, which reveals that the initial years of GLORYS2V3 might experience some problems. The first 10 years in GLORYS2V3 are quite suspicious because of its large deviation from the other products. Such large differences should be noticeable in the heat content changes or surface fluxes. Nevertheless, after 2007 all the reanalyses time series agree well and the OGCM hindcast deviates from the reanalyses. It is noteworthy that the observations improve considerably around that period due to an increasing number of Argo floats in use (Riser et al., 2016).

The reanalyses products used here are greatly influenced by the number of available in-situ observations. We further assess the sources of differences in the next section.

## 3.2 Source of Disparity

In order to further understand the difference between the AMET estimated from each atmosphere reanalyses product, we compare each component of AMET separately. We investigate the difference between each component of AMET at $60°$N estimated from ERA-Interim against those from MERRA2 and JRA55. It is noticed that the differences mainly originate from meridional temperature transport($vc_pT$) and geopotential energy transport ($vgz$). A simple linear regression shows the correlation between the difference of total energy transport and the difference of meridional temperature transport, taking ERA-interim and MERRA2, is 0.55, while for ERA-Interim and JRA55 that is very small. In addition, the correlation between the difference of total energy transport and the difference of geopotential energy transport ($vgz$), for ERA-Interim and MERRA2 is 0.56 and for ERA-Interim and JRA55 that is 0.60. For the other components, the correlations between them and the total difference are neglectable. The results are all obtained with a confidence interval over 95%. This is generally the case as large differences in temperature transport between reanalyses products are found at all latitudes (not shown). Such differences are consistent with the fact that the temperature transport and geopotential energy transport have larger contribution to the total AMET (see Figure 1). Note that the differences of each AMET component between every two products are of the same order of magnitude as the absolute values of that component. Besides, the latent heat transport agrees well between all the chosen atmospheric products, in terms of the mean and anomalies (not shown). A similar result was found by Dufour et al. (2016) in their study using more reanalyses data sets.

In order to know the relative contribution of each field to the difference of the total AMET among the chosen reanalyses, a direct comparison of the vertical profile of temperature and meridional velocity fields between ERA-Interim and MERRA2 is presented in Figure 5, as an example. We take the monthly mean temperature and velocity fields of ERA-Interim and MERRA2 from 1994 to 1998, in which the biggest difference was observed (Figure 3, taking into account the running mean of 5 years). For the sake of a point-wise comparison, the fields from MERRA2 are interpolated onto the vertical grid of ERA-Interim. This shows that these two reanalyses products differ substantially regarding each variable field (Figure 5a and b). Big differences in temperature reside mostly at the tropopause, while large differences in meridional wind component are distributed over the entire vertical column of the tropopause. Such differences in both fields are expected to be responsible for the difference in temperature transport ($vc_pT$). Large differences are found in geopotential height fields too (not shown). It should be noted that this comparison is carried out on pressure levels and the mass conservation is not ensured. Therefore it can only provide insight qualitatively and a quantitative contribution of the difference in each single field to the temperature transport can not be identified here.

Differences between every two chosen atmospheric products are found at nearly each pressure level. Given the data available, this analysis is not sufficient to explain conclusively where the uncertainty mainly comes from in terms of the dynamics and physics in the atmosphere model and data assimilation system. We do find that uncertainties as indicated by the spread between the datasets, in both the temperature and meridional velocity fields, are too large to constrain the AMET. Hence studies on low

frequency variability of energy transports and associated variables, should be interpreted with care as the reanalyses products differ substantially and we cannot make a priori judge how close they are to actual energy transports since independent direct observations are not available.

For the ocean, fortunately independent observations of OMET in the Atlantic Ocean are available. First, OMET estimated from ORAS4, GLORYS2V3, SODA3 and the OGCM hindcast is evaluated against OMET measured at $26.5°$N. Given in Figure 6, the inter-comparison shows that the reanalyses products capture roughly the mean amplitude of the OMET. Some large events are captured as well, such as the strong weakening in 2009. Statistically, the mean OMET provided by RAPID ARRAY is $1.21 \pm 0.27 PW$. It is higher than all the chosen products here. The mean OMET in ORAS4 is $0.66 \pm 0.27 PW$, in GLORYS2V3 it is $0.89 \pm 0.52 PW$, in SODA3 it is $0.81 \pm 0.52 PW$ and in OGCM hindcast is $1.05 \pm 0.21 PW$. This means that all chosen products underestimate the mean OMET at $26.5°$N in the Atlantic basin. Of all products, ORAS4 has the largest bias. The std of OMET given by ORAS4 is the same as that from RAPID ARRAY, while both in GLORYS2V3 and SODA3 we find a higher std of OMET. The OGCM hindcast has a relatively small OMET std of 0.21 PW. In terms of the correlation and standard deviation, ORAS4 and the OGCM hindcast agree well with observations. It is noteworthy that NEMO does not assimilate ocean data. The simulation is only constrained by the surface fluxes. To conclude, the heat transport at $26.5°$N is too low in these products.

Moreover, the comparison of time series in the chosen reanalyses and OSNAP observations is given in Figure 7. Due to the limited length of the OMET time series, only ORAS4 and SODA3 are included in the comparison. It can be noticed that the OMET given by ORAS4 is quite comparable to that in OSNAP in terms of the amplitude and variations. For most of the time within the observation period, OMET in ORAS4 falls into the range of the OSNAP observation including the uncertainty margins. The mean of OMET in ORAS4 is $0.39 \pm 0.11 PW$, which is quite similar to the mean OMET $0.45 \pm 0.07 PW$ of OSNAP. However, OMET in SODA3 has a larger mean and standard deviation than the OMET in OSNAP and thus deviates from the observation.

Just as in the atmosphere we would like to study the temperature and meridional current velocity contributions to the ocean heat transport to identify the sources of the difference between products. However, due to the nature of curvilinear grid, the comparison of local fields after interpolation is not trustworthy. To get further insight, we calculate the ocean heat content (OHC), since the convergence of the heat transports are likely related to OHC change. A full budget analysis was not feasible as most datasets did not include the surface fluxes. Figure 8 illustrates the OHC (Figure 8a) and the OHC anomalies (Figure 8b) quantified from ORAS4, GLORYS2V3, SODA3 and the OGCM hindcast. It depicts the OHC integrated in the polar cap (from $60°$N to $90°$N) over all depths. The mean OHC in ORAS4 is $4.48 \pm 0.78 * 10^{22} J$, in GLORYS2V3 is $4.23 \pm 0.59 * 10^{22} J$ and in SODA3 is $3.79 \pm 0.93 * 10^{22} J$, while the OGCM hindcast shows a much larger mean OHC of $7.85 \pm 0.58 * 10^{22} J$. The variations are similar between chosen products. Regarding the OHC anomalies in Figure 8b, a positive trend of OHC anomalies in the polar cap is captured by each product. However, the variations are different and these are reflected in the standard deviation of OHC anomalies time series. Qualitatively, the variations of OHC in the chosen reanalyses at polar cap can be taken as a sign of AA but a quantitative evaluation of AA is not possible due to large differences between products. To conclude, for the OHC there are large difference between chosen products while their variations agree very well. Since OHC is a function of

temperature fields only, this can imply that temperature profiles are different among all the chosen ocean reanalyses data sets. The chosen reanalyses data sets agree well. The differences of OHC between chosen products are partially consistent with the differences that we found for OMET. However, the OHC anomalies agree better with each other than the absolute OHC, which indicates that the trend of OHC is captured in a similar way among all the ocean reanalyses products.

## 3.3 MET and the Arctic

In previous sections it is found that MET of different reanalyses products at subpolar and subtropical latitudes differ substantially from each other. In order to further evaluate AMET and OMET given by different reanalyses and provide more insight, we investigate the links between MET and remote regions. We focus on the Arctic because previous studies indicate a strong role for subpolar MET in low frequency variability in the Arctic region. Given the complexity of the interaction between MET and the Arctic and the short time series available, determining cause-effect relations is out of scope for this paper. That is, we aim to compare the relation between MET and the Arctic within each reanalysis product to investigate the physical plausibility and compare it with previous studies that use data from one reanalysis product or from coupled climate models (e.g. Graversen (2006); Van der Swaluw et al. (2007); Graversen et al. (2008); Jungclaus and Koenigk (2010); Kapsch et al. (2013)).

Many of these studies perform linear regressions between a time series of MET and gridpoint values of other physical variables. Here we follow the same procedure and perform linear regressions of sea level pressure (SLP), 2 meter temperature (T2M) and sea ice concentration (SIC) anomalies on AMET and OMET anomalies at $60°$N for all the chosen products. We show linear regressions in summer and winter separately in order to account for the seasonal variability. We do note that correlations are higher when focusing on a particular season than a whole year. It should also be noted that there are strong trends in OMET, T2M and SIC. We removed them by applying a polynomial fit to the time series on each grid point. We find that the second order polynomial fit is able to capture the trend without losing variations at interannual time scales. Hereafter we only address detrended OMET, T2M and SIC. For the sake of consistency, the regressions are carried out on the surface fields included in each respective reanalyses product. For instance, the regression of SLP on AMET estimated from ERA-Interim, involves SLP fields from ERA-Interim itself. For the ocean reanalyses, as they all apply forcing derived from ERA-Interim, the regressions are performed on the fields from ERA-Interim. Note that there is a known issue with the quality of sea ice field close to the north pole in ERA-Interim, which can be inferred from an evaluation of reanalyses data sets concerning near surface fields in Lindsay et al. (2014). Following the regressions performed by Van der Swaluw et al. (2007) and Jungclaus and Koenigk (2010), we repeated the same procedure here with AMET at interannual scales.

First we investigate the links between MET and the Arctic in winter. The regressions of anomalies of multiple fields on AMET anomalies at $60°$N in each atmospheric product in winter are shown in Figure 9. The regression coefficients reach maximum when the regressions are instantaneous with given fields. In ERA-Interim and JRA55, AMET is correlated with SLP over the Greenland, the North Atlantic, the Barents Sea, the Kara Sea and the northern part of Eurasian continent. It suggests that an increase in subpolar AMET is linked to a northward advection over the Greenland which can bring relatively warm and humid air into the Arctic. Such patterns are consistent with the relatively warm air over the Greenland and part of the Central Arctic close to the Eurasian side shown in Figure 9d and f. Using ERA-40, Graversen (2006) found similar correlation between

AMET and surface air temperature (SAT) at the Greenland Sea and Barents Sea as Figure 9d and f, without time lag. This is also consistent with a model study by Jungclaus and Koenigk (2010). The reducing of sea ice concentration with increasing AMET at the Baffin Bay and the northern part of Barents Sea given by Figure 9g and i is consistent with the relations between AMET and T2M. A further eddy decomposition of AMET following the method from Peixoto and Oort (1992) indicates that heat

transported by standing eddies has the biggest contribution to the total AMET (not shown), which is consistent with Graversen and Burtu (2016). These patterns are found only in ERA-Interim and JRA55, but not in MERRA2. Given the difference in AMET amongst products, MERRA2 provides an entirely different story about AMET and the statistical relation with subpolar and Arctic atmospheric circulation. Hence, there is also large uncertainty in the assertion that heat and humidity transport by stationary eddies contribute to the changes in the subpolar and Arctic regions at interannual time scales.

Moreover, similar to Van der Swaluw et al. (2007) and Jungclaus and Koenigk (2010), we investigate the links between the variability of OMET and variations of multiple fields at interannual ($\sim 5$ year) time scales. The regressions of anomalies of multiple fields on detrended OMET anomalies at 60°N in winter are shown in Figure 10 with OMET leading by 1 month. The regression coefficients are maximal when the OMET leads by 1 month. In ORAS4 and SODA3, increasing OMET can lead to a decrease in SLP in the Arctic, while in ORAS4 this polar-low is much stronger. This indicates that an increase in OMET

is related to warm and humid air transport over the North Atlantic. Such patterns explain the correlation between OMET and T2M at the Greenland Sea and Barents Sea in Figure 10d and f, as well as the anticorrelation between OMET and SIC in the same regions in Figure 10g and i. Meanwhile, GLORYS2V3 tells an entirely different story. This is mainly due to the difference between OMET in this dataset compared to the other ocean data sets during the 1990s as shown in Figure 4.

In general, reduction of OMET leads to an increase in the growth rate of SIC, which is consistent with studies performed

with global climate models at decadal to inter-decadal time scales (e.g. Van der Swaluw et al. (2007); Jungclaus and Koenigk (2010); van der Linden et al. (2016)). Studies with observations of sea ice at the Barents Sea and OMET across Barents Sea Opening (BSO) also confirm the strong correlation between the OMET and sea ice variation over the Barents Sea (Årthun et al., 2012; Onarheim et al., 2015). However, note that some discussed regions are below the significance of 95%.

In summer, the situation becomes more intricate and unclear. The instantaneous regressions of anomalies of multiple fields

on AMET anomalies at 60°N in each atmospheric product in summer are shown in Figure 11. A high pressure center in the central Arctic is linked to an increase in AMET in all products. However, large differences are found in the relations between AMET and T2M and SIC. Strong positive correlations between AMET and T2M are found over the Greenland in both ERA-Interim and JRA55, but not in MERRA2. However, anticorrelations between AMET and T2M are observed at the Barents Sea and the Kara Sea in ERA-Interim and MERRA2, but not in JRA55. Links between AMET and SIC differ much between chosen

products, they are consistent with relations between AMET and T2M in each individual product, through. Consequently, the consistency between surface fields and AMET between chosen products in summer is even worse compared to winter. Given the differences between chosen reanalyses and relatively low statistical significance, it is quite difficult to make inference about the relation between AMET and T2M and SIC in summer.

It can be noticed that the consistency of associations between AMET and multiple fields is better in winter than that in

summer within the chosen products. Atmospheric dynamical processes are more dominant in winter, which is also reflected in

large scale patterns of variability such as the AO and NAO which are more pronounced in winter than in summer (e.g. Lian and Cess (1977); Curry et al. (1995); Goosse et al. (2018)). Therefore the regressions of SLP, T2M and SIC on AMET in winter are easier to understand than those in summer.

Similar issues are found in the regressions of the same fields on OMET at 60°N in each oceanic reanalysis product in summer, which are shown in Figure 12 with OMET leads by 1 month. Regarding the relations between OMET and SLP, a dipole pattern is observed in each oceanic reanalysis dataset, but the patterns are different in ORAS4 and SODA3 compared to those in GLORYS2V3. Different relations between OMET and T2M are found among all the products. In all the chosen oceanic reanalyses data sets, SLP and T2M are weakly correlated with OMET compared to those in winter. Although strong correlations between SIC and OMET are found in each oceanic reanalysis product (Figure 12g, h and i), the patterns are not consistent among them. Note that the statistical significance in these regressions are very low.

In this section we compared the reanalysis data with findings from previous studies. We found that ERA-Interim and JRA55 are most consistent with the results given by coupled numerical models in winter, while MERRA2 does not corroborate model studies. For the ocean, results from ORAS4 and SODA3 are more consistent with literature in winter. The regressions of anomalies from multiple fields on AMET and OMET anomalies in winter are easier to understand than those in summer. However, given the low statistical significance and the difference among chosen products, it is still hard to determine which atmospheric product provides a more convincing plausible interannual variations in AMET.

## 4 Discussion

In this study we found substantial differences between reanalyses products. In order to improve the accuracy of variability of AMET and OMET estimated from reanalyses, one needs more observations to constrain the models. Vertical profiles differ substantially between products and surface and top of the atmosphere radiation budget are too uncertain to constrain variability in the different products. Climate models already provide information on the interaction between atmosphere and ocean and connections provided by the energy transport from mid to high latitudes (Shaffrey and Sutton, 2006; Van der Swaluw et al., 2007; Jungclaus and Koenigk, 2010). This can potentially sketch the mechanism of the interaction between energy transport and the Arctic climate change. Moreover, some studies point out that the latent heat is more influential on the Arctic sea ice rather than the dry static energy (Kapsch et al., 2013; Graversen and Burtu, 2016). With improved reanalyses products and independent observations, such as ocean mooring arrays and atmospheric in-situ and remote observations, to validate the reanalyses, the validity of these mechanisms can be further studied.

The regression of SIC on OMET suggests that sea ice variation is sensitive to changes of meridional energy transport at subpolar latitudes, which is noticed by other studies on SIC and MET as well (Van der Swaluw et al., 2007; Jungclaus and Koenigk, 2010; van der Linden et al., 2016). ORAS4 and SODA3 show a large anticorrelation between SIC and OMET in winter around Greenland Sea and Barents Sea. However, GLORYS2V3 does not show this relation. The differences in OMET are reflected in the regressions on sea ice. The strong connection between OMET from mid-to-high latitudes and the Arctic sea ice indicates an indirect link between midlatitudes and the Arctic. Many studies that explored these remote links

found large scale "horseshoe" and dipole patterns over the Atlantic (Czaja and Frankignoul, 2002; Gastineau and Frankignoul, 2015; Delworth et al., 2017). However, the physical mechanism remains disputable. Overland et al. (2015) and Overland (2016) propose that the multiple linkages between the Arctic and midlatitudes are based on the amplification of existing jet stream wave patterns, which might also be driven by tropical and midlatitudes SST anomalies (Screen and Francis, 2016; Svendsen et al., 2018). Cohen et al. (2014) lists possible pathways for the teleconnection between the Arctic and midlatitudes, including changes in storm tracks, the jet stream, and planetary waves and their associated energy propagation. However, due to the shortness of time series, a small signal-to-noise ratio, uncertain external forcing, and the internal atmospheric variability (Overland, 2016; Barnes and Screen, 2015), this question has no easy answer.

Previous studies have shown that the variations of total OMET are very sensitive to the changes of its overturning component (e.g. McCarthy et al. (2015); Lozier et al. (2019)). Hence, AMOC can serve as a indicator of the changes of OMET. In our case, a quantitative estimation of the difference in AMOC among the chosen datasets is beyond our scope. However, the downward trend of AMOC, which has been reported by several studies (Smeed et al., 2014; McCarthy et al., 2015; Oltmanns et al., 2018), is consistent the downward trend observed in OMET at $60°$N in our chosen oceanic reanalyses (see Figure 4). After visiting six oceanic reanalyses data sets, Karspeck et al. (2017) find the reanalyses products are not consistent in their year-to-year AMOC variations. The discrepancy between AMOC represented by each reanalyses product may explain the difference in OMET in each reanalysis dataset.

## 5 Conclusions

This study aimed to quantify and inter-compare AMET and OMET variability from 3 atmospheric and 3 oceanic reanalyses data sets at subpolar latitudes. It also serves to illustrate the relation between AMET and OMET with high latitude climate characteristics. The study is motivated by previous studies with coupled models that show a strong relation between meridional energy transport and sea ice. It is also motivated by previous studies with reanalysis data, where generally only one reanalysis data set is considered, and which includes mostly only oceanic or atmospheric analysis.

All selected data sets agree on the mean AMET and OMET in the Northern Hemisphere. The results are consistent with those achieved over the previous 20 years (Trenberth and Caron, 2001; Fasullo and Trenberth, 2008; Mayer and Haimberger, 2012). However, when it comes to anomalies at interannual time scales they differ from each other, both spatially and temporally. Although there is overlap of observational data assimilated by different reanalyses products, large deviations still exist in main fields, especially for the vertical profiles of temperature and velocity in atmospheric reanalyses. Some reanalyses quality reports (Simmons et al., 2014, 2017; Uotila et al., 2018) have raised warnings for the use of certain variables from reanalyses. A further investigation of the relations between multiple fields in the Arctic and meridional energy transport shows that the Arctic climate is sensitive to the variations of AMET and OMET in winter. The patterns in ERA-Interim and JRA55 are more consistent in winter. For the ocean, ORAS4 and SODA3 provide similar patterns in winter. Based on our results, it seems that AMET and OMET cannot be constrained by the available observations. The reanalyses data sets are not designed for the studies on energy transport, specifically. The existence of sources and sinks in reanalyses data sets introduces large uncertainties in

the computation of energy transport (Trenberth, 1991; Trenberth and Solomon, 1994). As a consequence, much care should be taken when adopting the reanalyses for investigations on energy balance and energy transport related issues, especially for the ones aiming at relatively large time scales.

*Author contributions.* Y. Liu, J. Attema and W. Hazeleger designed this study, performed computations using reanalyses and analyzed the results. B. Moat performed OGCM simulation and contributed to the analysis.

*Competing interests.* The authors declare no competing interests.

*Acknowledgements.* The research was supported by the Netherlands eScience Center, Wageningen University, the National Oceanography Center in UK, and Blue Action project (European Union's Horizon 2020 research and innovation programme, grant number: 727852). The high resolution NEMO ORCA hindcast was complete in the project North Atlantic Climate System: Integrated Study (ACSIS) [grant number NE/N018044/1]. The authors are grateful for the high performance computational infrastructure (HPC cloud and Cartesius) provided by SURFsara in the Netherlands. We would like to express our gratitude to all the researchers working on the reanalyses data sets and making the data open to the public. We also want to thank the OSNAP (Overturning in the Subpolar North Atlantic Program) project and the RAPID-WATCH MOC monitoring project for making the observation data in the North Atlantic freely available.

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

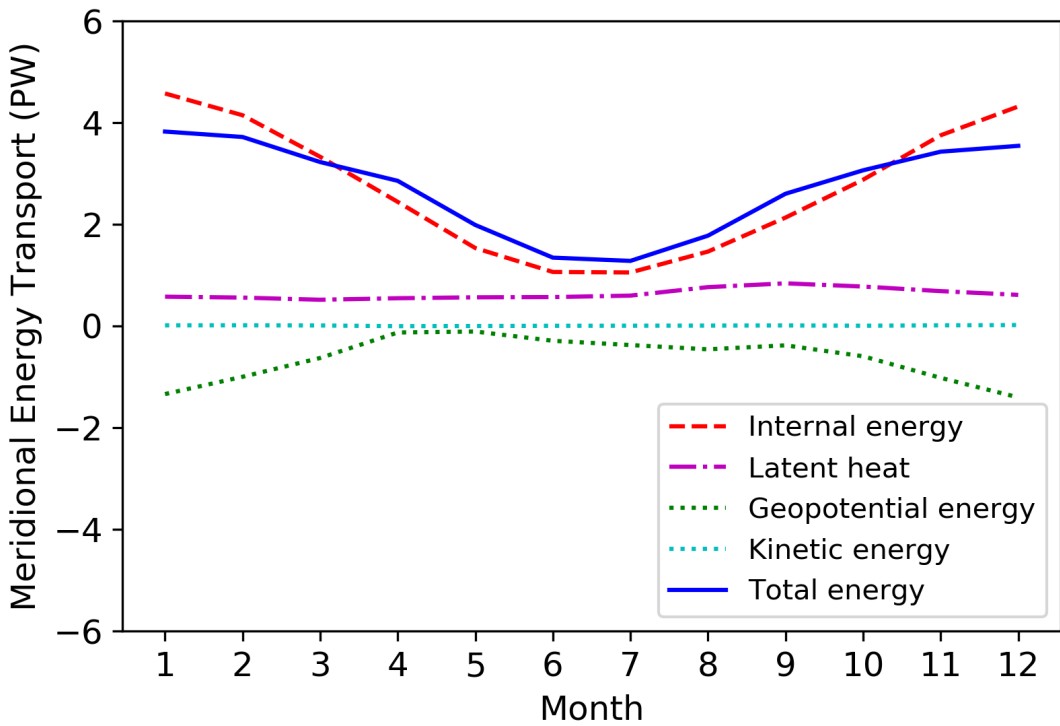

**Figure 1.** Estimation of mean AMET and each component in each month at $60°$N with ERA-Interim from 1979 to 2017. The unit is Peta Watt (PW).

**Table 1.** Basic specification of reanalyses products included in this study

| Type | Product Name | Producer | Period | Temporal Resolution | Spatial Resolution / Grid |
|------|------|------|------|------|------|
| | ERA-Interim | ECMWF | 1979 - 2017 | 6-hourly | TL255, L60 up to 0.1 hPa |
| Atmosphere | MERRA2 | NASA | 1980 - 2017 | 3-hourly | $0.5°$ x $0.625°$, L72 up to 0.01 hPa |
| | JRA55 | JMA | 1979 - 2016 | 6-hourly | TL319, L60 up to 0.1hPa |
| | ORAS4 | ECMWF | 1979 - 2016 | Monthly | ORCA1 |
| Ocean | GLORYS2V3 | Mercator-Ocean | 1993 - 2014 | Monthly | ORCA025 |
| | SODA3 | Univ. of Maryland | 1980 - 2014 | 5-daily | MOM5 |

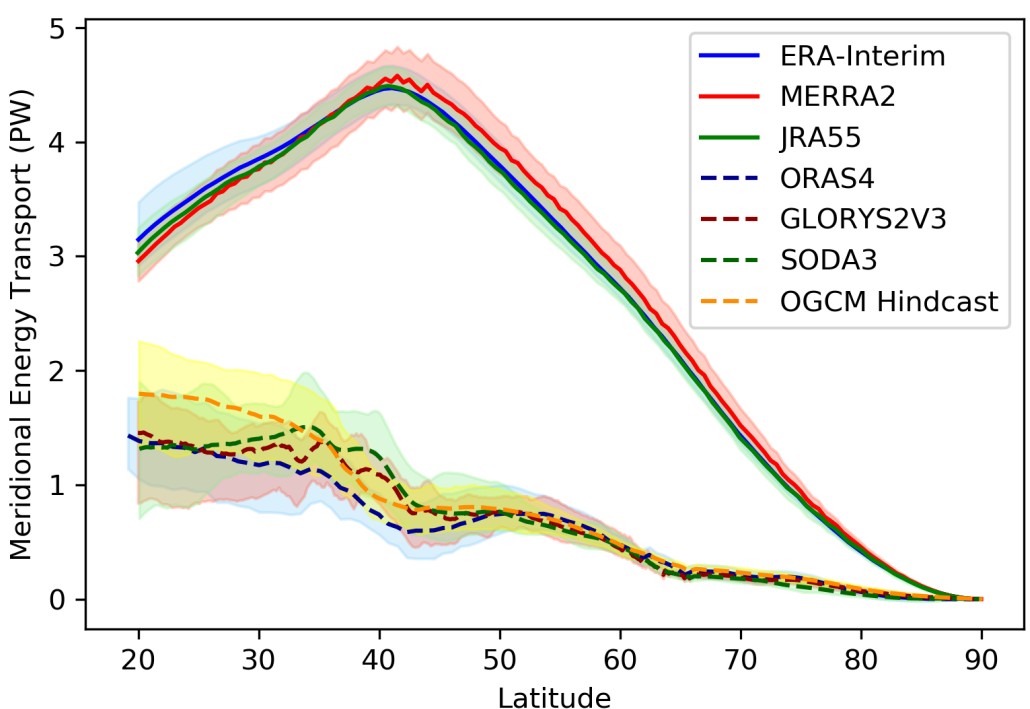

**Figure 2.** Mean AMET and OMET over the entire time span of each product as function of latitude in the Northern Hemisphere. AMET are illustrated with solid lines while OMET with dash lines. The shades represent the full range of MET across the entire time series at each latitude. The time span of each product used in this study is given in Table 1.

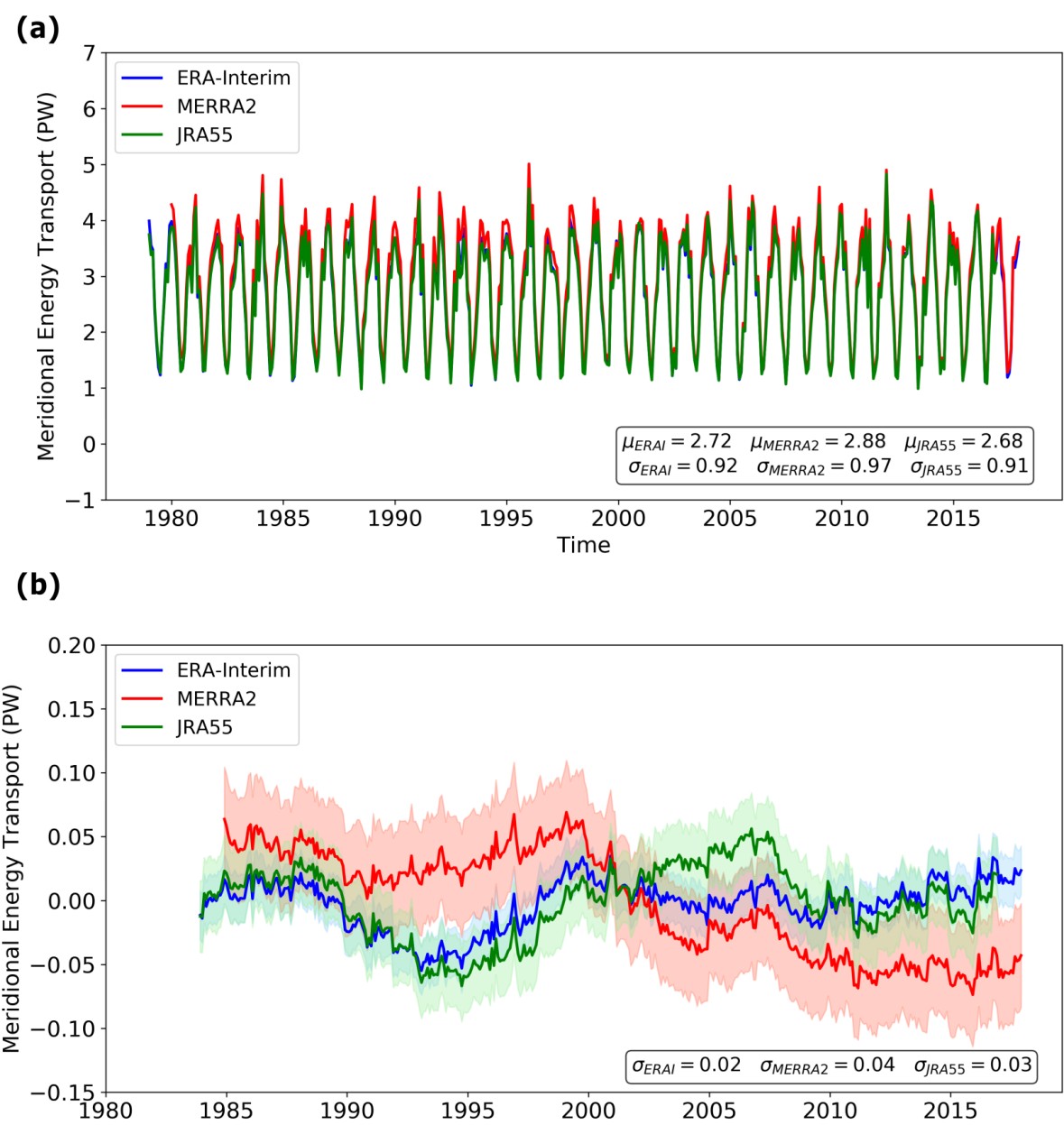

**Figure 3.** Time series of zonal integral of AMET at $60°$N without/with low pass filter. (a) The original time series and (b) the ones with low pass filter include signals from ERA-Interim (blue), MERRA2 (red) and JRA55 (green). For the low pass filtered ones, we take a running mean of 5 years. The shades represent the confidence intervals with one standard deviation. $\sigma$ is the standard deviation and $\mu$ is the mean of the entire time series.

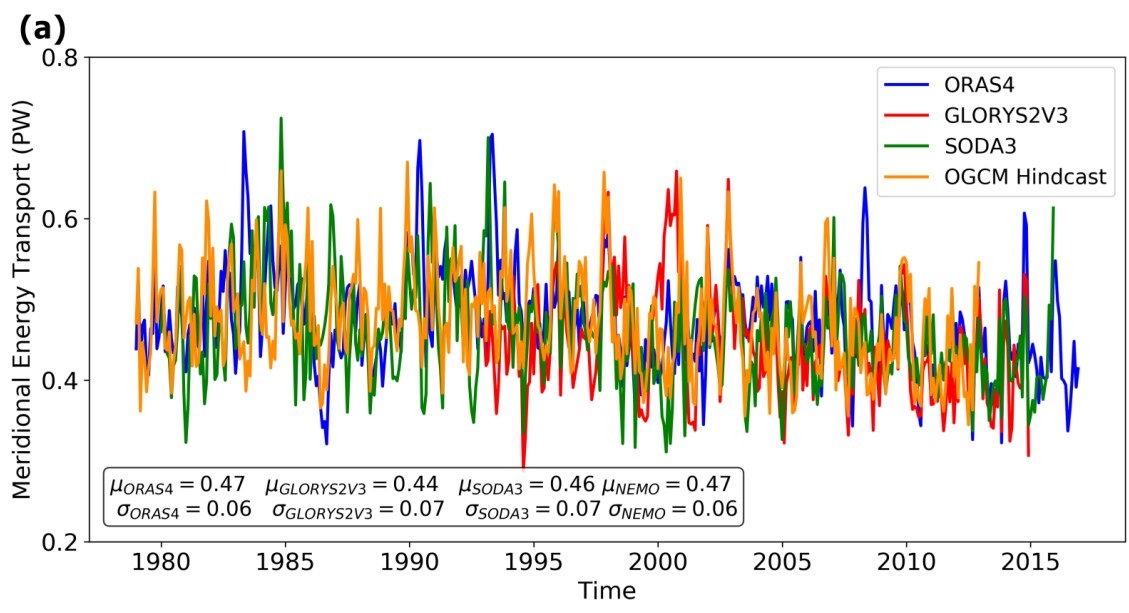

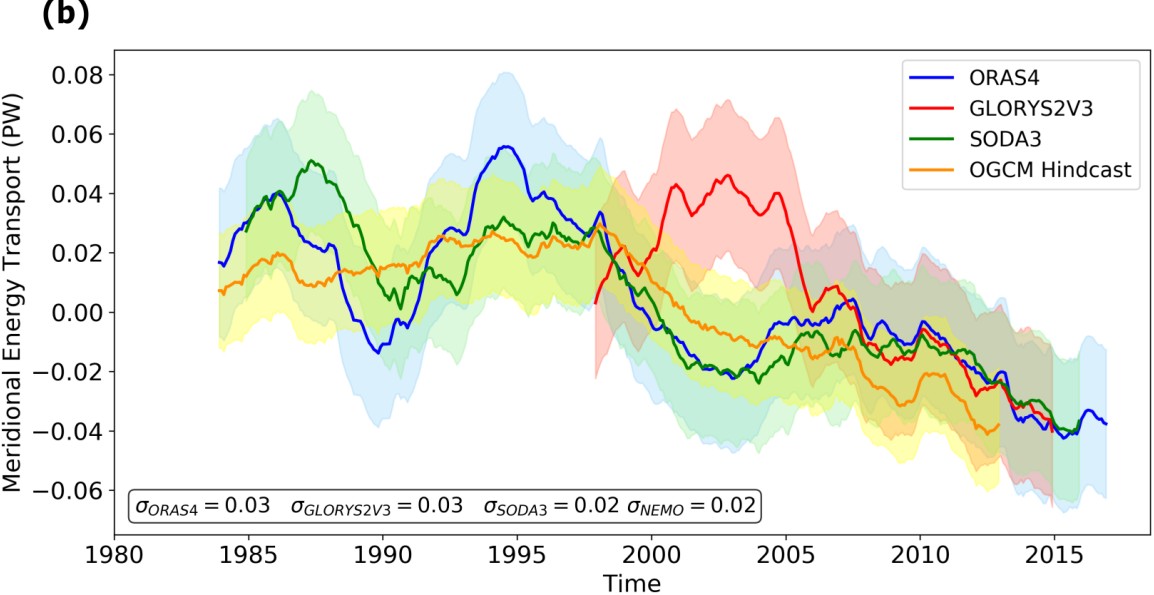

**Figure 4.** Time series of zonal integral of OMET at $60°$N without/with low pass filter. (a) The original time series and (b) the ones with low pass filter include signals from ORAS4 (blue), GLORYS2v3 (red), SODA3 (green) and the OGCM hindcast (yellow). For the low pass filtered ones, we take a running mean of 5 years. The shades represent the confidence intervals with one standard deviation. $\sigma$ is the standard deviation and $\mu$ is the mean of the entire time series.

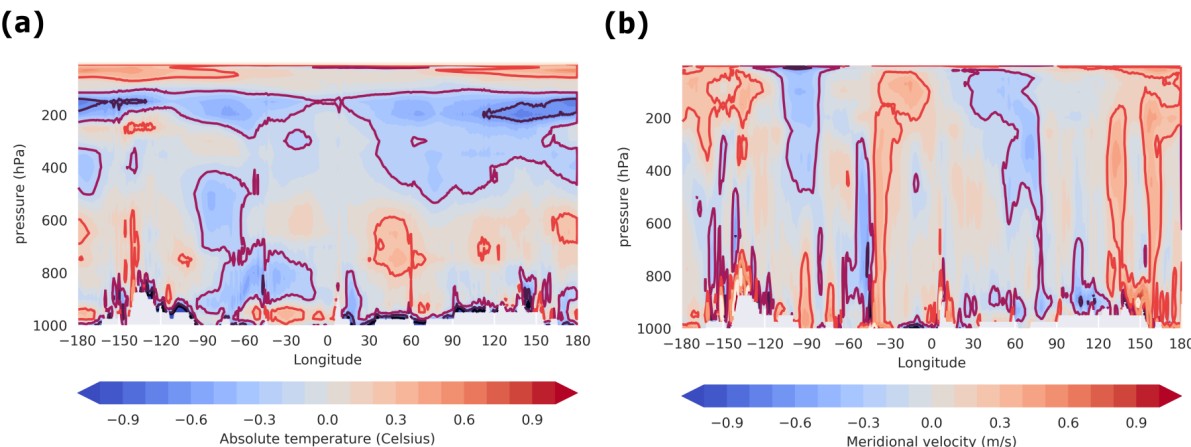

**Figure 5.** Difference in temperature, meridional wind velocity and temperature transport between MERRA2 and ERA-Interim at $60°$N. The vertical profile of (a) temperature difference and (b) meridional wind velocity difference are calculated from the climatology of each fields from 1994 to 1998, respectively.

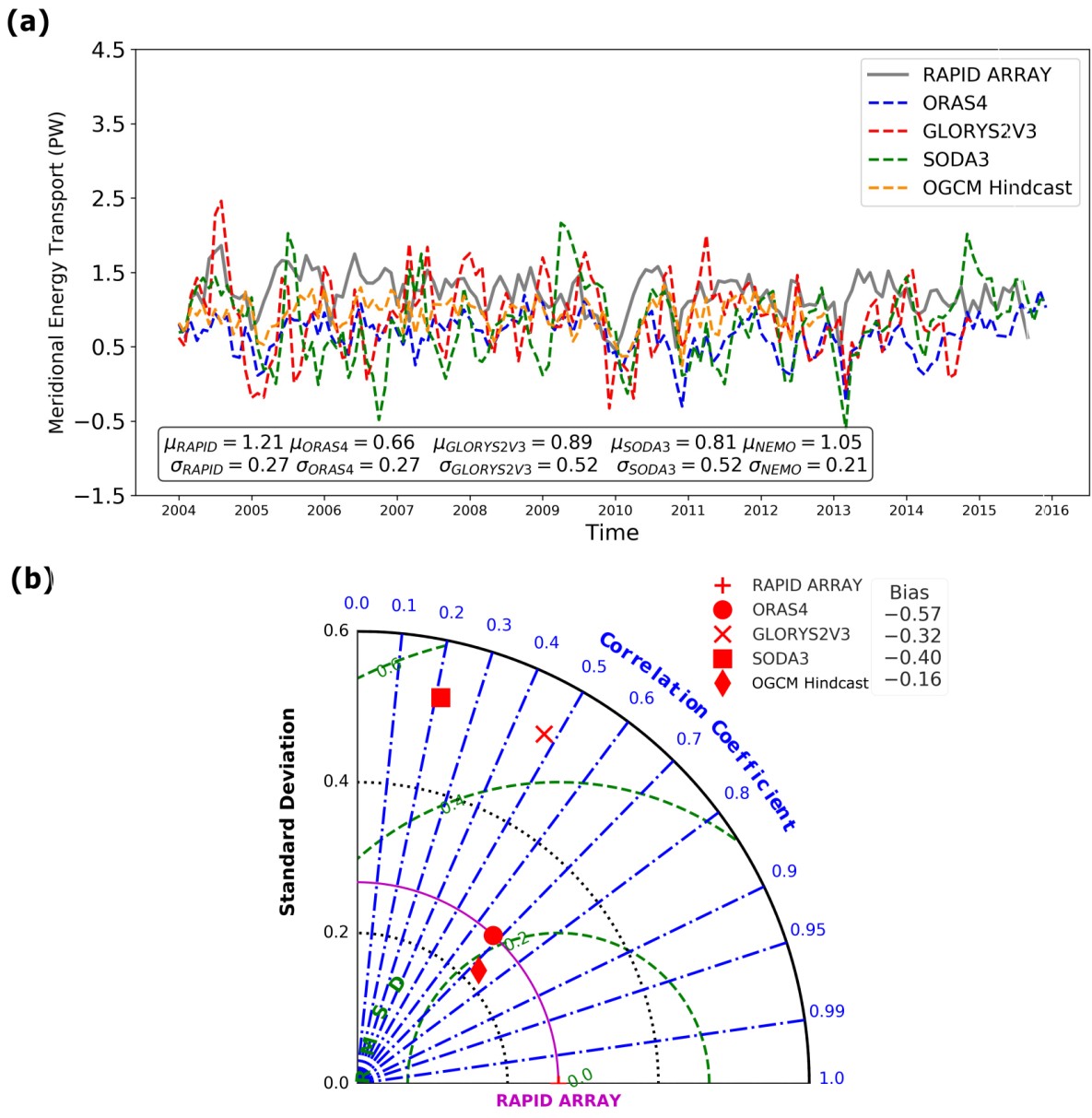

**Figure 6.** OMET estimated from ORAS4 (blue), GLORYS2V3 (red), SODA3 (green) and the OGCM hindcast (orange) compared to the RAPID ARRAY observation (gray) at 26.5°N across the Atlantic basin. The time series of OMET is presented in (a). The statistical properties are shown in (b) Taylor Diagram, including bias, correlation (blue), standard deviation (black) and root mean square deviation (green). $\sigma$ is the standard deviation and $\mu$ is the mean of the entire time series.

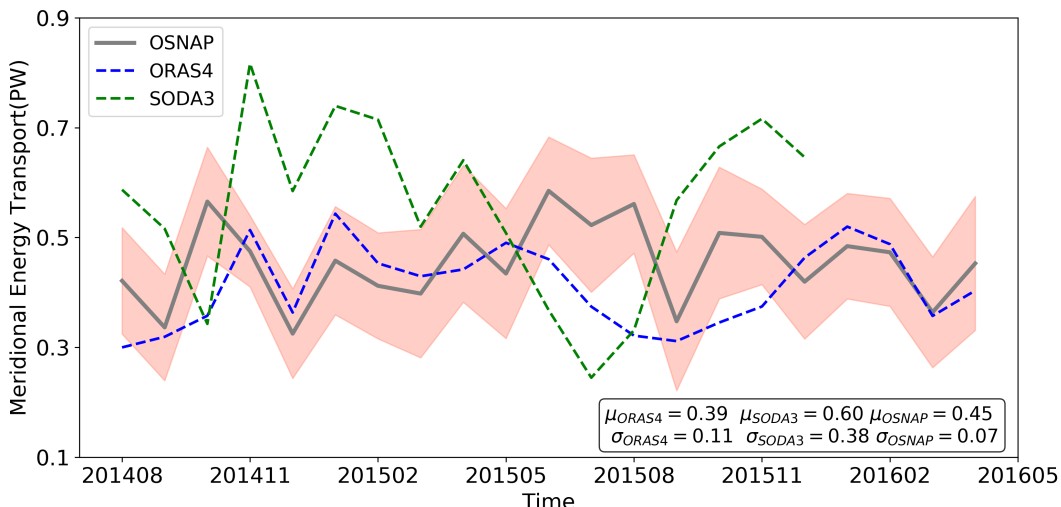

**Figure 7.** OMET estimated from ORAS4 (blue), SODA3 (green) and compared to the OSNAP observation (gray) at subpolar Atlantic basin. The range of uncertainty from OSNAP observation is marked by the red shade. $\sigma$ is the standard deviation and $\mu$ is the mean of the entire time series.

**(a)**

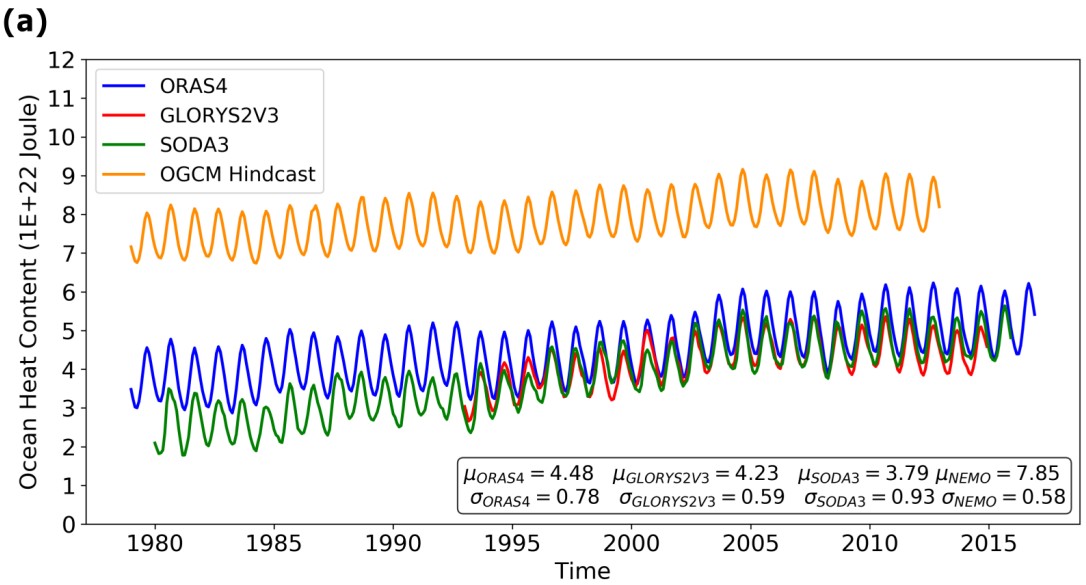

**(b)**

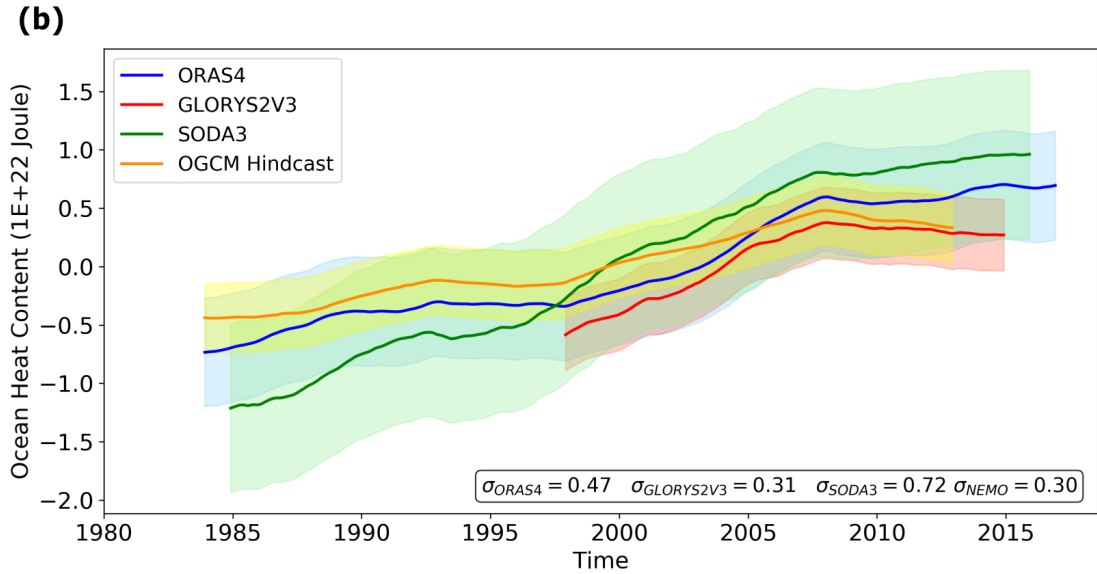

**Figure 8.** Time series of (a) ocean heat content (OHC) and (b) OHC anomalies with a low pass filter at the polar cap. The OHC is integrated from surface to the bottom between 60°N and 90°N. It is estimated from ORAS4 (blue), GLORYS2V3 (red), SODA3 (green) and the OGCM hindcast (yellow). The shades represent the confidence intervals with one standard deviation. $\sigma$ is the standard deviation and $\mu$ is the mean of the entire time series.

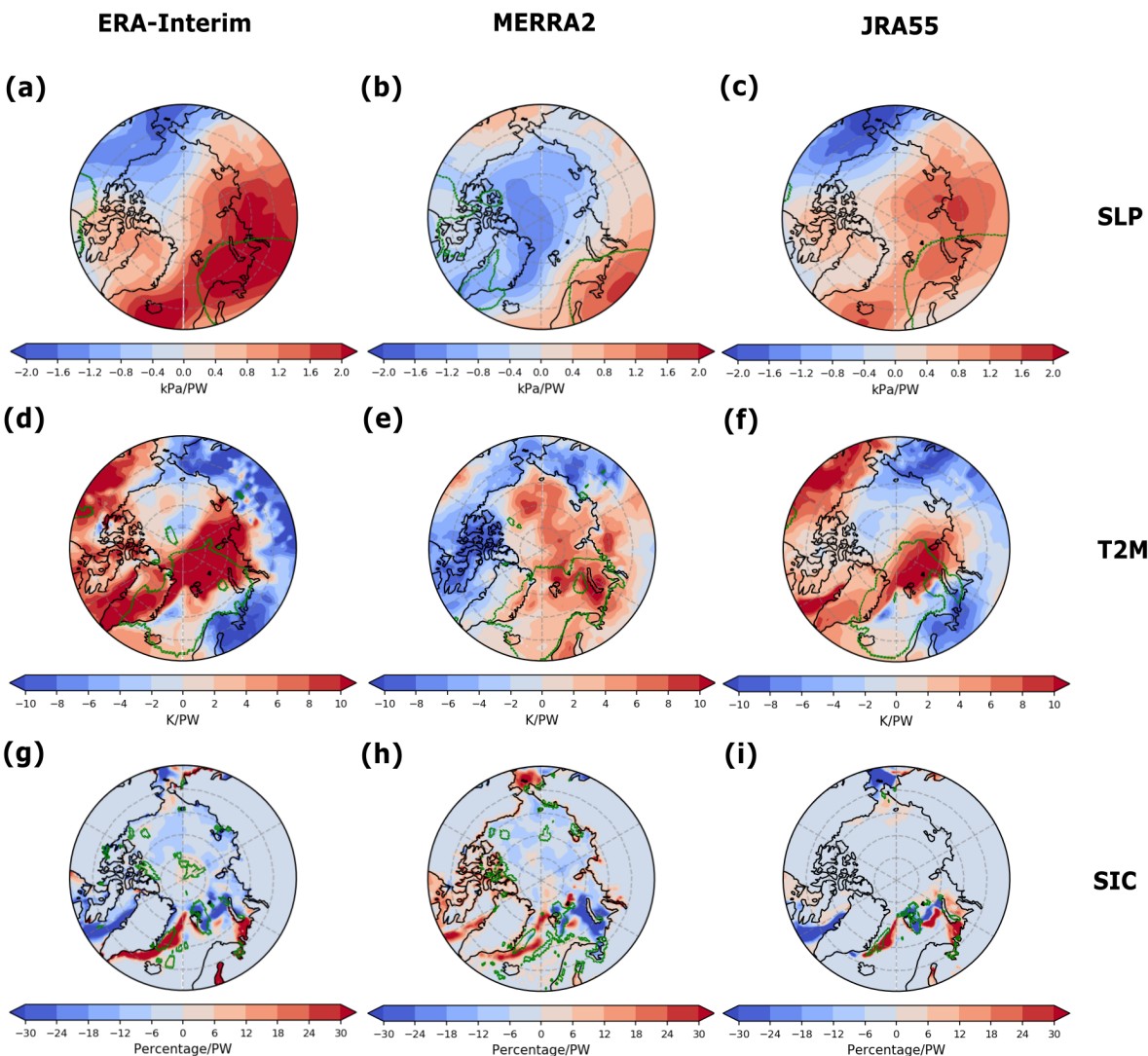

**Figure 9.** Regressions of sea level pressure, 2 meter temperature and sea ice concentration anomalies on AMET anomalies at $60°$N in winter (DJF) at interannual time scales with no time lag. The monthly mean fields are used here after taking a running mean of 5 year. Both the 2 meter temperature and sea ice concentration are detrended. From left to right, they are the regressions on AMET of (a, d, g) ERA-Interim, (b, e, h) MERRA2 and (c, f, i) JRA55. The green contour lines indicate a significance level of 95%.

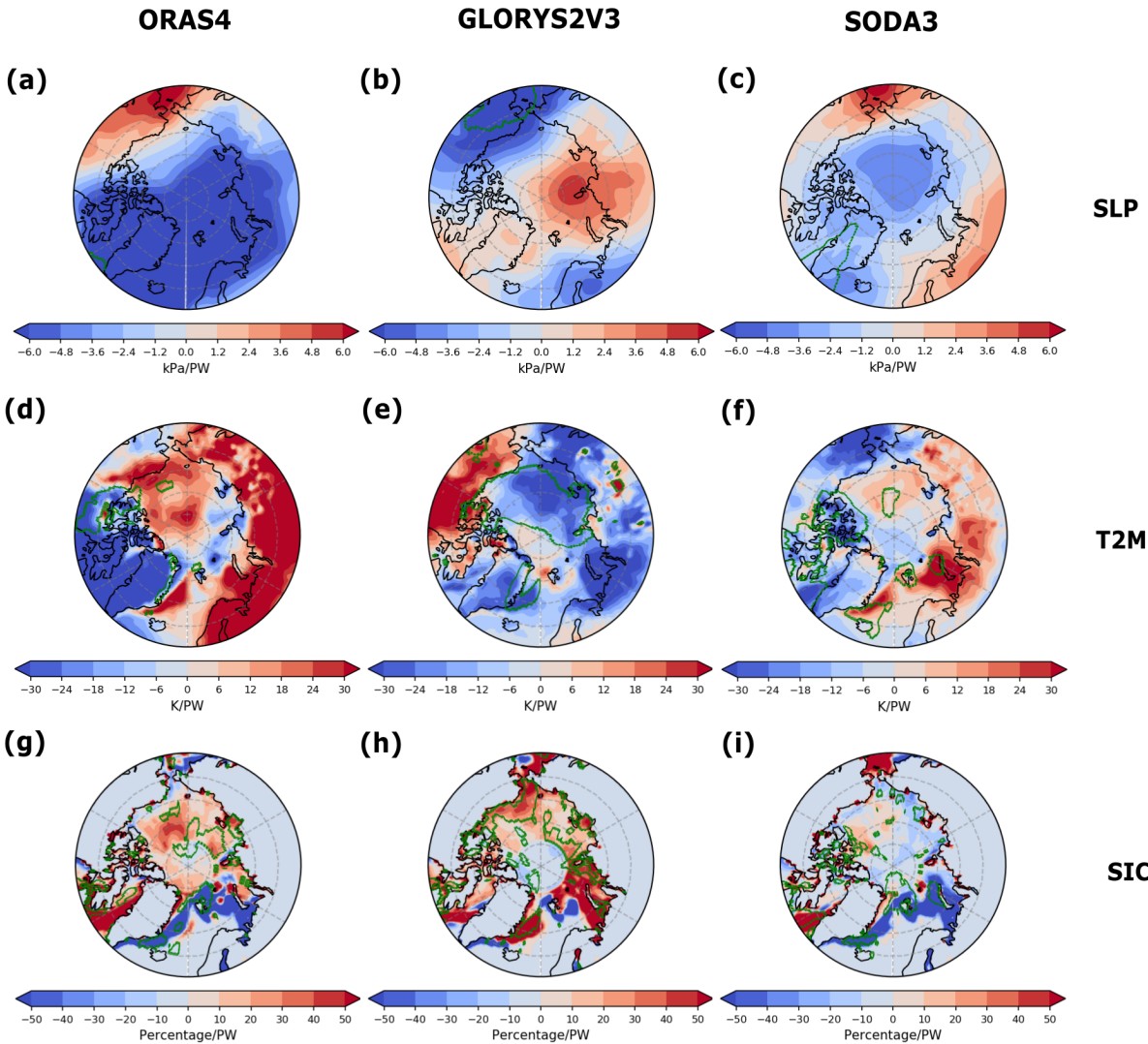

**Figure 10.** Regressions of sea level pressure, 2 meter temperature and sea ice concentration anomalies on OMET anomalies at 60°N in winter (DJF) at interannual time scales. OMET leads the fields by one month. The 2 meter temperature, sea ice concentration and OMET are detrended. From left to right, they are the regressions on OMET of (a, d, g) ORAS4, (b, e, h) GLORYS2V3 and (c, f, i) SODA3. The green contour lines indicate a significance level of 95%.

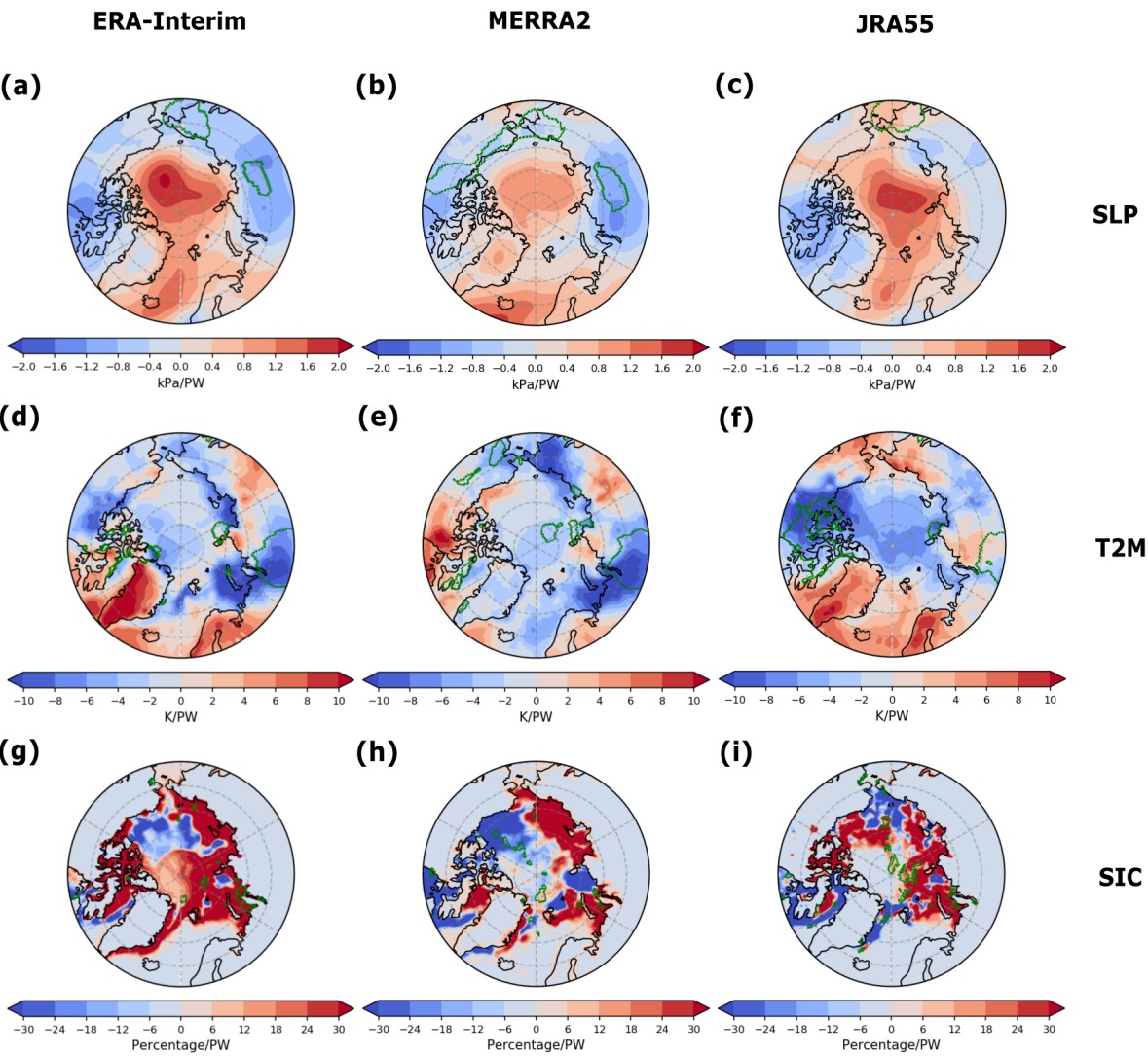

**Figure 11.** Regressions of sea level pressure, 2 meter temperature and sea ice concentration anomalies on AMET anomalies at 60°N in summer (JJA) at interannual time scales with no time lag. The monthly mean fields are used here after taking a running mean of 5 year. Both the 2 meter temperature and sea ice concentration are detrended. From left to right, they are the regressions on AMET of (a, d, g) ERA-Interim, (b, e, h) MERRA2 and (c, f, i) JRA55. The green contour lines indicate a significance level of 95%.

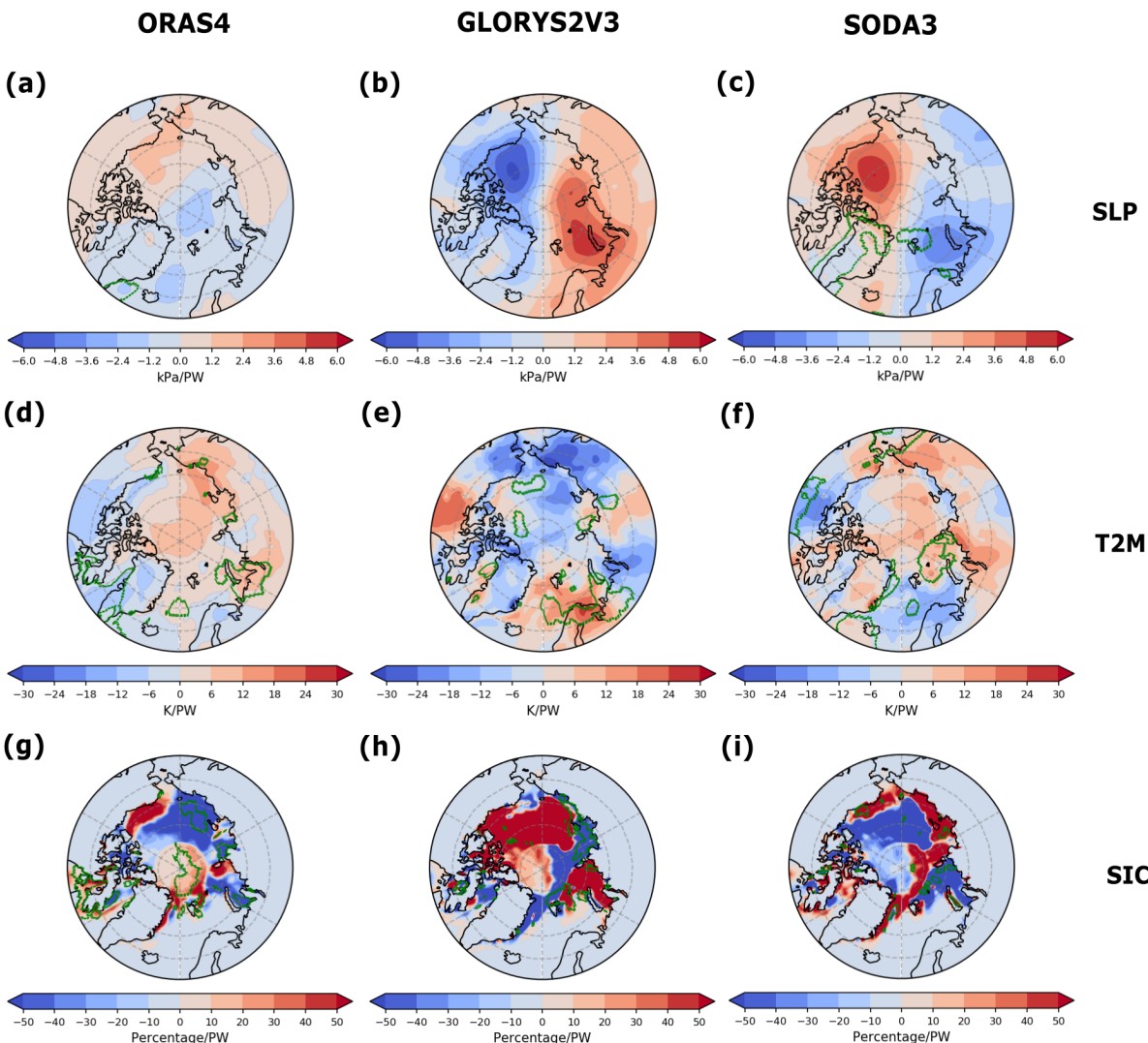

**Figure 12.** Regressions of sea level pressure, 2 meter temperature and sea ice concentration anomalies on OMET anomalies at 60°N in summer (JJA) at interannual time scales. OMET leads the fields by one month. The 2 meter temperature, sea ice concentration and OMET are detrended. From left to right, they are the regression on OMET of (a, d, g) ORAS4, (b, e, h) GLORYS2V3 and (c, f, i) SODA3. The green contour lines indicate a significance level of 95%.