# Peer review of "Synthesis and evaluation of historical meridional heat transport from midlatitudes towards the Arctic"

_Earth System Dynamics, 2019_

## Referee Comment (RC1) · Anonymous Referee #1 · 5 Jun 2019

Review of Synthesis and evaluation of historical meridional heat transport from midlatitudes towards the Arctic by Liu et al

The present study is concerned with the inter-annual to decadal variability of poleward atmospheric and oceanic heat transports. For this purpose, the authors computed these quantities from three atmospheric and three oceanic reanalysis products, respectively. They find marked discrepancies in the results from the different products, which leads them to the conclusion that the current generation of reanalyses are of limited use for investigation of low-frequency variability in the derived fluxes. The layout of the study is good and its results are of interest to the community. However, it appears there exist severe technical shortcomings in the computation of the atmospheric transports (which I will describe in the more specific comments), which puts into question all

the subsequent results and conclusions. I therefore encourage the authors to carefully check their chain of computations and revise their results. I consider this a necessary precondition for publication of the present study. More specific comments are in the following.

Major concerns: 1) Figure 3a indicates that the annual cycle of AMET from ERA-Interim and JRA55 is very different, with yearly minima in JRA55 going down to ∼0.8PW,while annual minima in ERA-Interim remain closer to 1.5PW, which implies a huge relative discrepancy (>50%) in the amplitude of the annual cycle of AMET from the two products. From our own computations using basically identical scripts for both products I can tell that their annual cycle of AMET actually agrees very well (within <10%). Also, low-pass-filtered variability of AMET looks quite different from our results compared to the authors' figure 3b. Thus, I presume there is something wrong in the authors' computation of AMET (at least for ERA-Interim and JRA55, I cannot judge the results from MERRA2), possibly in the mass adjustment that they apply. I recommend to thoroughly check the chain of computations. NCAR provides a quite detailed step-by-step instruction how to perform these computations: http://www.cgd.ucar.edu/cas/catalog/newbudgets/index.html

2) The authors note that there exist improved diagnostic equations for energy budget diagnostics (Mayer et al. 2017; Trenberth and Fasullo 2018), but do not use those. I strongly recommend to make use of these updated equations. There is no reason not to do so.

3) Ocean energy budget (section 2.3.2) and discussion about reference temperature: The authors discuss the need for a reference temperature as long as the mass budget is not closed. This is indeed important and this issue has been extensively discussed in the oceanographic literature, most notably Schauer and Beszczynska-Möller (2009). However, the present study only considers oceanic transports in a zonally integrated sense. Full zonal cross-sections should have a net mass flux close to zero, making the use of a reference temperature unnecessary. The same applies for the statement

about recirculation (p8 l13). In fact, for zonal integrals, there only is a small imbalance coming from P-E, leaving a small ambiguity, which in the same manner applies to the atmosphere. For that reason, the same reference temperature should be used for both atmosphere and ocean to obtain consistent results (Mayer et al. 2017). The discussion on these issues must be clarified.

Minor comments:

Generally: the plural of "reanalysis" is "reanalyses", while the plural of "reanalysis data set" is "reanalysis data sets". Please correct throughout the manuscript.

There are many inaccurate formulations throughout the manuscript. I picked only a selection in the following, but I generally recommend to carefully revise the manuscript in order to make it more concise.

P2L16: Is this result based on models? Please clarify.

P2L24: In this context it might be worth mentioning that ocean reanalyses do not show a clear sign of Arctic amplification in Arctic OHC increases (Mayer et al. 2016; von Schuckmann et al. 2018)

P2L34: Please rewrite the sentence to something like: "These are representations of the historical state of the atmosphere and ocean optimally combining available observations and numerical simulations using data assimilation techniques."

P3L3: Please spell out this acronym (and all others).

P3L7: Please be more specific about the "model". Is this a forced ocean model run?

P3L23: "higher" than what?

P3L24: change "preferably" to "preferable"

P3L25: "For an inter-comparison purpose, they better not resemble each other". What is meant here exactly? Please reword.

P4L7-8: Is there a reference for the statement about divergent winds? It might be worth checking Graversen et al. (2007)

P4L14: add "scheme" after "assimilation".

P4L22: add "upper air" before "observations"

P4L26: oceans → ocean's

P4L30: data with 3D-Var assimilation → analyses with a 3D-Var FGAT assimilation scheme

P5L9: Not quite right. The forcing is a combination of ERA-Interim fluxes (e.g. short-wave radiation) and bulk formulae using ERA-Interim near-surface parameters. Please correct.

P5L16-17: "To be consistent with the other two reanalyses datasets assessed in this study, the SODA 3.4.1 is chosen since it applies surface forcing from ERA-Interim". This statement seems to be opposite of what you say above in P3L25

P5L18: r –> R

P6L5: What is the "Drakkar forcing data" based on?

P7L21: explain u_c and v_c

P8L10: the equation gives OHC across a certain circle of latitude. Is this meaningful? How would that relate to the transports across that latitude? Did the authors mean OHC integrated across the area north of a given latitude?

P8L11-12: I can reassure you that ocean reanalyses do have sources and sinks from temperature increments, but they do not suffer from mass inconsistencies as atmospheric reanalyses do. The divergence of ocean currents exactly balances the surface freshwater flux and local sea level variations, so there is no mass adjustment needed. Please see the NEMO documentation for details (Madec 2008).

P9L3: Do you mean "decorrelation"?

P9L6: Do you mean "by a factor of 3"?

P9L8: Why not do this the other way round? Apply filter first, and then estimate effective degrees of freedom.

P9L10: I do not understand the statement about statistical significance

P9L18: not only solar radiation, but also OLR. In that sense, transports balance NET radiation.

P9L26: How do you know this?

P9L29: "ERA-Interim res" is not shown in the figure.

P9L21: 1.21PW is probably too high (see major comment #1)

P10L10: Here it is important to know whether you are using monthly or sub-monthly data, which should be stated in the methods section. If you use monthly means, you will miss eddy transports and consequently underestimate ocean heat transports.

P10L12: What is plotted in Fig4? The poleward component of OMET? How is this obtained, if OMET is computed on the tripolar NEMO grids?

P10L19: Is "hindcast" the appropriate term? Isn't it a forced model run?

P11L1: correlation or regression?

P11L5: r=0.07 seems very small (again, see major comment #1)

P11L20: With these diagnostics, you are running into problems with reference temperature, as the point-wise transports obviously do not have a zero mass flux. Figure 7d (showing T_mean*delta_v) will have much larger values when you use K instead of C, leading to a different conclusion. Which one would be correct? I suggest to remove 7c and 7d and discuss a and b in more detail. Instead of 7c and d, it would be interesting to show the difference sections also for JRA55.

P12L5: Please use a different name than "NEMO", as all your ocean products are based on NEMO.

P12L15ff: Similar to my above comment: I do not understand why you look at bands of OHC and not at OHC north of 60N.

P13L9: Better use an independent SIC product. SIC from ERA-Interim is of questionable quality, which can be seen e.g. from the "disc" around the North Pole. Also, ORAS4 does not have an active sea ice model. Would you expect a correlation between OMET and SIC then?

P13L18: no -> not

P13L32: Fig 13: I am not sure what is shown here. Are these instantaneous regressions? The legend says 1-month lag - does this make sense when using 12-monthly smoothed data? At which lag do you get highest correlation?

P14L1: I think one has to be careful with the timescales here. What timescales are the cited studies looking at?

P14L25: How can you see this from the time series?

P14L27: Be cautious: A lot of heat transported across 60N is stored in the North Atlantic or released from there through air-sea fluxes and will never reach sea-ice covered regions.

P14L28: "patterns" of what?

P14L31: remove "the" before "tropical"

P15L8: "less consistent" than what?

P15L24-26: Hard to understand. Please rewrite the sentence.

References:

Graversen, R. G., E. Källén, M. Tjernström, and H. Körnich, 2007: Atmospheric

mass‐transport inconsistencies in the ERA‐40 reanalysis. Q. J. R. Meteorol. Soc., 133, 673–680.

Madec, G., 2008: NEMO ocean engine: Note du Pôle de modélisation de l'Institut Pierre-Simon Laplace No 27. Tech. rep,.

Mayer, M., L. Haimberger, M. Pietschnig, and A. Storto, 2016: Facets of Arctic energy accumulation based on observations and reanalyses 2000–2015. Geophys. Res. Lett., 43.

——, ——, J. M. Edwards, and P. Hyder, 2017: Toward Consistent Diagnostics of the Coupled Atmosphere and Ocean Energy Budgets. J. Clim., 30, 9225–9246.

Schauer, U., and A. Beszczynska-Möller, 2009: Problems with estimation and interpretation of oceanic heat transport–conceptual remarks for the case of Fram Strait in the Arctic Ocean. Ocean Sci., 5, 487–494.

von Schuckmann, K., and Coauthors, 2018: Copernicus Marine Service Ocean State Report 2nd issue. J. Oper. Oceanogr., 11, S1–S142, https://doi.org/10.1080/1755876X.2018.1489208.

Trenberth, K. E., and J. T. Fasullo, 2018: Applications of an updated atmospheric energetics formulation. J. Clim., 6263–6279, https://doi.org/10.1175/JCLI-D-17-0838.1.

---

## Referee Comment (RC2) · Anonymous Referee #2 · 4 Jul 2019

How much of the variability and recent decline of Arctic sea ice can be attributed to local processes, versus energy import by the ocean or atmosphere, is an important question. One problem is that energy-consistent coupled climate models tend to have large biases in the region, while it is hard to get good observations of integrated energy transport due to the extensive coverage required. This paper examines the latter approach, by comparing the meridional energy transports and their low-frequency variability in different reanalyses datasets in the atmosphere and ocean. It then briefly addresses the impacts on Arctic climate.

This topic is important, and the paper potentially very useful in helping if, and which dataset to use to study these questions; however it has a number of problems : mostly the computation of the atmospheric transport seems flawed except in one case, and

the last section (impact on the Arctic) is not enough developed to be useful. I therefore recommend major revisions to address these issues.

Major points :

(A) Computation of atmospheric transport (AMET)

Computation of an energy transport from reanalysis can be difficult due to lack of mass /energy conservation, sampling problems, or numerical schemes different from the original. It seems from figure 2 that MERRA2 and JRA55 suffer from large errors ("noise" on the curves) of the order of O.1-0.2 PW for the long-term mean.

Unfortunately, this means these products will be unusable to study the variability of the transport. Indeed, it is evident on Fig. 3 that the interannual variability at 60°N is of the same order as this long-term noise.

There are 2 possibilities :

- This problem comes from the reanalysis themselves : the conclusion is then that only ERA-interim is useable. This would be a useful result in itself; but the study of the differences between ERA and the other reanalyses becomes pointless and the paper could instead concentrate more on the impacts on the Arctic.

- This is a problem in the calculations from the reanalysis data, which then has to be solved, and the other results corrected accordingly.

(B) Impacts on the Arctic (section 3.3) The problem of this section is that it gives a few quick examples of regressions of characteristics of the Arctic climate on energy transports, but that they are too short to be really useful. - Show at least the winter and summer seasons, as results could be quite different. - Why show SLP and tempera-ture for AMET, but sea-ice for OMET ? Why not show the same variables for both for comparison ? (at least sea ice and slp).

Other remarks : - p.3 line 24 : "it is preferable that they incorporate the latest..."

- p.3 line 25 : "they better not resemble each other" well, you certainly hope that different reanalyses would be consistent with each other !

- section 2.3.1 : what is the value of Lv used here ? Lv varies with T in nature, but not necessarily in models... Not sure what's used in reanalyses.

- p.7 : there may also be issues due to different horizontal advection schemes used in the reanalysis and in the post-treatment.

- p.8, line 6 : unit should be J/(kg°C) or J.°C-1.kg-1.

- p.8 : reference temperature. If the unit of potential temperature Theta is in °C, then substracting a reference temp. of 0°C does not accomplish anything. Are you instead converting Theta from K to °C to avoid cancellation of large terms problems ? This is very unclear.

- p.9, l25 : The differences in resolution are actually small. There must be another explanation to these variations, which are key (main point A)

- p.10, l10 : In ocean models that are not eddy-resolving, there is both an eddy-advection (Gent-McWilliams) and a diffusive heat transport, which can be significant compared to the resolved one. How were these incorporated in these analyses ? They absolutely need to be taken into account.

- section 3.2 and the accompanying figures for the atmosphere is a bit pointless given the low quality except for ERA-interim.

- p11, bottom : Are we looking in this section at the total OMET, or only the Atlantic OMET ? It would probably be more interesting to look at Atlantic only at 60°N (in line with section 3.3), although knowing the relative roles could be good. Same question for page 13.

- p12 : any idea about why ORAS4 seems to work best ?

- p13, l28 : why 5-years timescale ? Is it specific to the Arctic ? Are the regressions on

Fig.13 for 5-yr filtered data, or just year-to-year ?

- Figure 1 :this is hardly commented in the text: seasonal cycle, low contribution of LH (especially to the seasonal cycle)... By the way, I would replace this figure by either the components of the time-mean transport as a function of latitude, or by the mean seasonal cycle at 60°N of the different components. (With figures for the interannual variance if needed)

- Figure 2: The dashed-line is built using annual means ?

- Figure 3 : panel b) needs confidence intervals, based on interannual variance. For panel a), do the standard deviations include the seasonal cycle or are they for interannual anomalies ?

- Figure 4 : not sure what is the point of these figures, apart from showing that the high-res analyses have eddies ? It would be easier to copare maps integrated to the same low resolution, or time-means, and also to see the GM components.

- Figure 6 could easily be replaced by the figures given in the text. Note that latent heat transport may not contribute much to the differences because it's low to begin with, but also because it's concentrated in the near-surface layers, so does not suffer too much from slight mass flux imbalances.

- Figure 11, caption : I guess it is "interannual" time-scale ? (i.e. year-to-year variability)

- Figures 11-13 : consistent time-scales and variables, with different seasons would be nicer. Also, the green shading masks the color underneath, making it hard to read. Same-color shading maybe ?

- Figure 13b : there are strange-looking colors (opposite the rest of the Arctic...) near the pole in all 3 plots (abrupts changes of sign...) Is this an artefact ?

---

## Author Response (AR1)

Dear Editor Prof. Dr. Lohmann,

Thanks a lot for your time and the effort you put on processing and reviewing our paper. We write this letter to inform you the submission of revised version of our paper based on the remarks from two anonymous reviewers.

Besides, regarding your question about "it is very hard to include all NEMO components derived from ORAS4 to close the budget", we mean that some extra term enthalpy (e.g. heat transport related to eddy induced velocity) cannot be included as those fields are not saved.

Again, we would like to thank you for reviewing our work and finding helpful reviewers. We look forward to hearing from you about the progress of the following review procedure. We are sincerely gratitude to everything you made to improve this paper.

With best regards

Yang Liu, Jisk Attema, Ben Moat, and Wilco Hazeleger

Dear anonymous reviewer #1,

We are very glad to receive so many constructive comments and we would like to thank the reviewer for his help. We are able to address all points and will do our best to improve the quality of our paper accordingly. Below is our response to the reviewer's comments point by point:

 (the *original comments* are given by *Italic gray text*, and each follows our response in plain text.)

**Response to the major comments**

*1) Figure 3a indicates that the annual cycle of AMET from ERA-Interim and JRA55 is very different, with yearly minima in JRA55 going down to ~0.8PW,while annual minima in ERA-Interim remain closer to 1.5PW, which implies a huge relative discrepancy (>50%) in the amplitude of the annual cycle of AMET from the two products. From our own computations using basically identical scripts for both products I can tell that their annual cycle of AMET actually agrees very well (within <10%). Also, low-pass-filtered variability of AMET looks quite different from our results compared to the authors' figure 3b. Thus, I presume there is something wrong in the authors' computation of AMET (at least for ERA-Interim and JRA55, I cannot judge the results from MERRA2), possibly in the mass adjustment that they apply. I recommend to thoroughly check the chain of computations. NCAR provides a quite detailed step-by-step instruction how to perform these computations: http://www.cgd.ucar.edu/cas/catalog/newbudgets/index.html*

After thoroughly checking our script and the whole computation procedure, we find that the discrepancy between AMET from ERA-Interim and JRA55 comes from the numerical scheme used in performing the mass correction. The barotropic mass correction method involves calculations of the divergence, the inverse Laplacian and a gradient (Trenberth, 1991). We used a finite difference (central scheme) method to compute these terms. However, after some tests we noticed that it is more accurate to bring the fields to the spectral domain and compute these terms via spherical harmonics. The adjustments to the barotropic wind as results of mass budget correction based on these two numerical methods differ much in the polar regions, in particular the amplitude. Since AMETs are very sensitive to the mass budget, the results with these two different numerical schemes can lead to very different results. After recomputing the AMET fluxes from ERA-Interim, MERRA2 and JRA55 with mass correction using spherical harmonics, now the difference between annual cycles of AMET from ERA-Interim and JRA55 is very small. The results are shown in the figure below. They are consistent with the reviewer's results. In our revised paper we will use the new numerical method for the diagnostics and update the results, figures and discussion.

[Figure]

The low frequency series of AMET from ERA-Interim, MERRA2 and JRA55 also changed due to the implementation of mass correction via computation with spherical harmonics. But as they are still very different, the major results in our paper remains the same. Changes are made to the regressions of different variables with AMET mainly.

Trenberth, K. E. (1991). Climate diagnostics from global analyses: Conservation of mass in ECMWF analyses. Journal of Climate, 4(7), 707-722.

*2) The authors note that there exist improved diagnostic equations for energy budget diagnostics (Mayer et al. 2017; Trenberth and Fasullo 2018), but do not use those. I strongly recommend to make use of these updated equations. There is no reason not to do so.*

Thanks for pointing out this. At the time the initial work for this manuscript was done these improved methods were not published yet. Now, together with the implementation of mass correction via spherical harmonics, we recomputed the AMET using the improved diagnostic equations for energy budget diagnostics (Equation 24 in Mayer et. al., 2017).

Mayer, M., Haimberger, L., Edwards, J. M., & Hyder, P. (2017). Toward consistent diagnostics of the coupled atmosphere and ocean energy budgets. Journal of Climate, 30(22), 9225-9246.

*3) Ocean energy budget (section 2.3.2) and discussion about reference temperature: The authors discuss the need for a reference temperature as long as the mass budget is not closed. This is indeed important and this issue has been extensively discussed in the oceanographic literature, most notably Schauer and Beszczynska-Möller (2009). However, the present study only considers oceanic transports in a zonally integrated sense. Full zonal cross-sections should have a net mass flux close to zero, making the use of a reference temperature unnecessary. The same applies for the statement about recirculation (p8 l13). In fact, for zonal integrals, there only is a small imbalance coming from P-E, leaving a small ambiguity, which in the same manner applies to the atmosphere. For that reason, the same reference temperature should be used for both atmosphere and ocean to obtain consistent results (Mayer et al. 2017). The discussion on these issues must be clarified.*

We agree with the reviewer that the reference temperature is not needed when the mass is perfectly balanced. In oceanographic literature it is common to use a reference temperature when calculating OMET in both observations and model diagnostics (e.g. Bryan, 1962; Hall and Bryden, 1982; Johns et. al., 2011; van der Linden et. al. 2019). As we cannot perform barotropic correction for the ocean, we take reference temperature as a compromise. In the ocean, with its strong boundary circulations even the smallest imbalance can lead to large errors in the heat flux. Besides, it is very hard to include all NEMO components derived from ORAS4 to close the budget. Thus, we think it is still safe to take a reference temperature. However, we agree with the reviewer that when taking the zonal integral the benefit from a reference temperature is not as large as when considering a single strait transports (Schauer and Beszczynska-Möller, 2009). To make the formulation much clear, we will modify our explanation about the usage of the reference temperature as well as the statement about recirculation. We will add Schauer and Beszczynska-Möller (2009) to our reference list and discuss the mass imbalance coming from P-E (Mayer et. al., 2017).

Bryan, K. (1962). Measurements of meridional heat transport by ocean currents. Journal of Geophysical Research, 67(9), 3403-3414.

Hall, M.M. and H. L. Bryden (1982) Direct estimates and mechanisms of ocean heat transport, Deep-Sea Research, Vol. 29, No. 3A, pp. 339 to 359

Johns, W. E., Baringer, M. O., Beal, L. M., Cunningham, S. A., Kanzow, T., Bryden, H. L., ... & Curry, R. (2011). Continuous, array-based estimates of Atlantic Ocean heat transport at 26.5 N. Journal of Climate, 24(10), 2429-2449.

Schauer, U., & Beszczynska-Möller, A. (2009). Problems with estimation and interpretation of oceanic heat transport–conceptual remarks for the case of Fram Strait in the Arctic Ocean. Ocean Science, 5(4), 487-494.

van der Linden, E. C., Le Bars, D., Bintanja, R., & Hazeleger, W. (2019). Oceanic heat transport into the Arctic under high and low CO2 forcing. Climate Dynamics, 1-18.

**Response to minor comments**

*Generally: the plural of "reanalysis" is "reanalyses", while the plural of "reanalysis dataset" is "reanalysis data sets". Please correct throughout the manuscript.*

We will correct these incorrect formulations.

*P2L16: Is this result based on models? Please clarify.*

Their study uses reanalyses data (Yang et. al., 2010). We will further specify the details of these studies that listed in this paragraph.

Yang, X. Y., Fyfe, J. C., & Flato, G. M. (2010). The role of poleward energy transport in Arctic temperature evolution. Geophysical Research Letters, 37(14).

*P2L24: In this context it might be worth mentioning that ocean reanalyses do not show a clear sign of Arctic amplification in Arctic OHC increases (Mayer et al. 2016; von Schuckmann et al. 2018)*

Thanks for the comment. We will include this point.

*P2L34: Please rewrite the sentence to something like: "These are representations of the historical state of the atmosphere and ocean optimally combining available observations and numerical simulations using data assimilation techniques."*

We will rewrite the sentence following your suggestion.

*P3L3: Please spell out this acronym (and all others).*

The acronym will be spelled out.

*P3L7: Please be more specific about the "model". Is this a forced ocean model run?*

Yes, this is an ocean model run forced by surface fields from the Drakkar Surface Forcing data set version 5.2.

*P3L23: "higher" than what?*

This is a bit confusing. Here we mean we choose reanalysis products with relatively high resolution (compared to old reanalyses) as the quantification of energy transport need this. We will change "higher" to "high", just to be accurate.

*P3L24: change "preferably" to "preferable"*

We will change it.

*P3L25: "For an inter-comparison purpose, they better not resemble each other". What is meant here exactly? Please reword.*

Thanks for pointing this out. The sentence is indeed very confusing. Here we want to emphasize that we try to include various reanalyses data sets to make the comparison more informative. We will delete this sentence.

*P4L7-8: Is there a reference for the statement about divergent winds? It might be worth checking Graversen et al. (2007)*

Berrisfold et. al. (2011) discussed the improvement of divergent winds in their work. We will include this reference now.

Berrisford, P., Kållberg, P., Kobayashi, S., Dee, D., Uppala, S., Simmons, A. J., ... & Sato, H. (2011). Atmospheric conservation properties in ERA‐Interim. Quarterly Journal of the Royal Meteorological Society, 137(659), 1381-1399.

*P4L14: add "scheme" after "assimilation".*

We will add it.

*P4L22: add "upper air" before "observations"*

We will add it.

We will correct it.

*P4L30: data with 3D-Var assimilation -> analyses with a 3D-Var FGAT assimilation scheme*

We will change it.

*P5L9: Not quite right. The forcing is a combination of ERA-Interim fluxes (e.g. shortwave radiation) and bulk formulae using ERA-Interim near-surface parameters. Please correct.*

Thanks for the information. We will correct it.

*P5L16-17: "To be consistent with the other two reanalyses datasets assessed in this study, the SODA 3.4.1 is chosen since it applies surface forcing from ERA-Interim". This statement seems to be opposite of what you say above in P3L25*

We will delete the apparently confusing statement in P3L25.

*P5L18: r —> R*

We will correct it.

*P6L5: What is the "Drakkar forcing data" based on?*

Thanks for reminding. The NEMO ORCA run is forced by the Drakkar Surface Forcing data set version 5.2, which supplies surface air temperature, winds, humidity, surface radiative heat fluxes and precipitation, and a formulation that parameterizes the turbulent surface heat fluxes and is provided for the period 1958 to 2012 (Dussin et al., 2014; Brodeau et al., 2010).

Brodeau, L., B. Barnier, A. M. Treguier, T. Penduff, and S. Gulev (2010), An ERA40-based atmospheric forcing for global ocean circulationmodels, Ocean Modell., 31, 88–104, doi:10.1016/j.ocemod.2009.10.005.

Dussin, R., B. Barnier, and L. Brodeau (2014), The making of Drakkar forcing set DFS5, DRAKKAR/MyOcean Rep. 05-10-14, LGGE, Grenoble,France.

*P7L21: explain u_c and v_c*

These are the correction terms for zonal and meridional wind components as a result of barotropic mass budget correction. We will further explain this based on the explanation given by Trenberth (1991).

Trenberth, K. E. (1991). Climate diagnostics from global analyses: Conservation of mass in ECMWF analyses. Journal of Climate, 4(7), 707-722.

*P8L10: the equation gives OHC across a certain circle of latitude. Is this meaningful? How would that relate to the transports across that latitude? Did the authors mean OHC integrated across the area north of a given latitude?*

The equation here is a bit confusing. We agree with the reviewer that OHC integrated in the polar cap is likely closely related to OMET at 60N (the chosen reanalysis products are all forced by surface fields from ECMWF atmospheric reanalyses.). We will update the equation here and calculate the OHC in the polar cap.

*P8L11-12: I can reassure you that ocean reanalyses do have sources and sinks from temperature increments, but they do not suffer from mass inconsistencies as atmospheric reanalyses do. The divergence of ocean currents exactly balances the surface freshwater flux and local sea level variations, so there is no mass adjustment needed. Please see the NEMO documentation for details (Madec 2008).*

We update this and put an accurate formulation now. However, it is very hard to include all NEMO components derived from ORAS4 to close the budget. See also van den Berk et. al. (2019) for the freshwater budget.

van den Berk, J., Drijfhout, S. S., & Hazeleger, W. (2019). Atlantic salinity budget in response to Northern and Southern Hemisphere ice sheet discharge. Climate Dynamics, 52(9-10), 5249-5267.

*P9L3: Do you mean "decorrelation"?*

Yes, this is what we aim to do.

*P9L6: Do you mean "by a factor of 3"?*

Yes.

*P9L8: Why not do this the other way around? Apply filter first, and then estimate effective degrees of freedom.*

Statistically, it seems to be more strict to estimate the effective degrees of freedom after applying filter. We will apply the filter first and then estimate effective degrees of freedom.

*P9L10: I do not understand the statement about statistical significance*

Here we mean that the relatively short records of reanalyses do not have many samples at interannual time scales, compared to outputs from numerical climate models. We will rephrase this to make it more clear.

*P9L18: not only solar radiation, but also OLR. In that sense, transports balance NET radiation.*

Thanks for the comment. We will correct this.

*P9L26: How do you know this?*

Now we notice that the latitudinal variation is due to the mass correction based on finite difference scheme. We will correct this.

*P9L29: "ERA-Interim res" is not shown in the figure.*

This is a mistake. We will correct this now.

*P9L21: 1.21PW is probably too high (see major comment #1)*

We will update this according to the results with improved methods. Now it is of the same scale as ERA-Interim.

*P10L10: Here it is important to know whether you are using monthly or sub-monthly data, which should be stated in the methods section. If you use monthly means, you will miss eddy transports and consequently underestimate ocean heat transports.*

With ORAS4 and GLORYS2V3, we use monthly data since only the data at monthly scales are available. For SODA3, we use sub-monthly data (5 day averaged) (which includes the eddy components). We will further explain this now after including your comment.

*P10L12: What is plotted in Fig4? The poleward component of OMET? How is this obtained, if OMET is computed on the tripolar NEMO grids?*

It is the poleward component of energy transport in the ocean. We regrid from curvilinear grid to lat-lon grid as we only want to emphasize the resolution and its potential influence on our results. But it seems to be a bit confusing, thus we decided to remove it. Thanks for asking.

*P10L19: Is "hindcast" the appropriate term? Isn't it a forced model run?*

The model is forced by (near) surface fields from historical data only. It is common to call this a hindcast in oceanographic literature (indeed, this differs from atmospheric literature). But we agree that it is also "a forced model run". We have added this description in the method section.

*P11L1: correlation or regression?*

We check the correlation between the total AMET and each component. So it is "correlation".

*P11L5: r=0.07 seems very small (again, see major comment #1)*

We will update this according to the new results.

*P11L20: With these diagnostics, you are running into problems with reference temperature, as the point-wise transports obviously do not have a zero mass flux. Figure 7d (showing T_mean*delta_v) will have much larger values when you use K instead of C, leading to a different conclusion. Which one would be correct? I suggest to remove 7c and 7d and discuss a and b in more detail. Instead of 7c and d, it would be interesting to show the difference sections also for JRA55.*

Thanks for the point. We agree that we don't have zero mass flux in this point-wise set-up. We will keep only Figure 7a and 7b.

The reason that we only show this for ERA-Interim and MERRA2 is that these two reanalyses have points at the same location in terms of pressure levels, lat and lon on the close-to-native grids that they are produced with (those grids are slightly interpolated grids compared with native grids, e.g. for ERA-Interim TL255 is roughly equal to 0.75 x 0.75 deg, and MERRA2 natively 0.5 x 0.625 deg). We only choose the points at exactly the same location. However, to include JRA55 (TL319 roughly equal to 0.5625 x 0.5625 deg) a further interpolation is inevitable, which can introduce errors. So we only make such point-wise comparison between MERRA2 and ERA-Interim, as a compromise.

*P12L5: Please use a different name than "NEMO", as all your ocean products are based on NEMO.*

We will change the name to "OGCM hindcast".

*P12L15ff: Similar to my above comment: I do not understand why you look at bands of OHC and not at OHC north of 60N.*

The explanation is given above. But we will switch to the OHC north of 60N.

*P13L9: Better use an independent SIC product. SIC from ERA-Interim is of questionable quality, which can be seen e.g. from the "disc" around the North Pole. Also, ORAS4 does not have an active sea ice model. Would you expect a correlation between OMET and SIC then?*

We wish to obtain a consistent picture by regressing OMET generated with models forced by ERA-Interim fields on sea ice also from ERA-Interim. Indeed, there is no active sea ice model, but for SIC the temperature criteria are assumed to be reasonable. Of course this is not the case for thickness or volume.

*P13L18: no -> not*

Corrected.

*P13L32: Fig 13: I am not sure what is shown here. Are these instantaneous regressions? The legend says 1-month lag - does this make sense when using 12-monthly smoothed data? At which lag do you get highest correlation?*

In this regression OMET leads the sea ice by one month. We observe the highest correlation between OMET and sea ice when OMET leads by one month. But the difference between correlations are very small, as long as OMET is leading.

*P14L1: I think one has to be careful with the timescales here. What timescales are the cited studies looking at?*

Thanks for the reminding. These studies focus on decadal to inter-decadal scales. We will add the timescales here.

*P14L25: How can you see this from the time series?*

The low frequency anomalies of OMET (Figure 5b) shows that GLORYS2V3 differs a lot to ORAS4 and SODA3. By saying this we want to emphasize that the differences in OMET are reflected in the regressions on sea ice. But this seems a bit confusing. We will rephrase this.

*P14L27: Be cautious: A lot of heat transported across 60N is stored in the North Atlantic or released from there through air-sea fluxes and will never reach sea-ice covered regions.*

Thanks for reminding. We will rephrase this part and mention that the relation is "indirect".

*P14L28: "patterns" of what?*

"Horse shoe" or "dipole" patterns of the link between OMET and SST over the Atlantic. We will add this to the text.

*P14L31: remove "the" before "tropical"*

We will correct this.

*P15L8: "less consistent" than what?*

Sorry for the typo. We mean they are not consistent with each other. We change "less consistent" to "not consistent".

*P15L24-26: Hard to understand. Please rewrite the sentence.*

We mean the sink and source in reanalyses cause large uncertainties in the computation of energy transport. We will rephrase this part.

Again, we'd like to express our gratitude to the time that the reviewer spent on it. Thank you very much!

With best regards

Yang Liu, Jisk Attema, Ben Moat, and Wilco Hazeleger

Dear anonymous reviewer #2,

First, we'd like to thank the reviewer for the time and effort on reviewing our paper. We manage to address all the points from the reviewer and try our best to improve the quality of our paper. Below is our response to the reviewer's comments point by point. We will revise the manuscript accordingly.

(the *original comments* are given by *Italic gray text*, and each follows our response in plain text.)

**Response to the major points**

*Computation of an energy transport from reanalysis can be difficult due to lack of mass / energy conservation, sampling problems, or numerical schemes different from the original. It seems from figure 2 that MERRA2 and JRA55 suffer from large errors ("noise" on the curves) of the order of O.1-0.2 PW for the long-term mean. Unfortunately, this means these products will be unusable to study the variability of the transport. Indeed, it is evident on Fig. 3 that the interannual variability at 60N is of the same order as this long-term noise.*

*There are 2 possibilities:*

*- This problem comes from the reanalysis themselves: the conclusion is then that only ERA-interim is useable. This would be a useful result in itself; but the study of the differences between ERA and the other reanalyses becomes pointless and the paper could instead concentrate more on the impacts on the Arctic.*

*- This is a problem in the calculations from the reanalysis data, which then has to be solved, and the other results corrected accordingly.*

Thanks for the comment. We also have concerns about the noisy annual mean AMET from MERRA2 and JRA55. After thoroughly checking our script and the whole computation procedure, we find that the "noise" in annual mean AMET from MERRA2 and JRA55 in Figure 2 comes from the numerical scheme used in performing the mass correction. The barotropic mass correction method involves calculations of the divergence, the inverse Laplacian and gradients (Trenberth, 1991). We used finite difference (central scheme) method to compute these terms. However, after some tests we noticed that it is more accurate to bring the fields to spectral domain and compute these terms using spherical harmonics. The adjustments to the barotropic wind as results of mass budget correction based on these two numerical methods differ much in the polar regions. Since AMET are very sensitive to the mass budget, the results with these two different numerical schemes can lead to very different results. After recomputing the AMET fluxes from ERA-Interim, MERRA2 and JRA55 with mass correction using spherical harmonics, now the noise in annual mean AMET has been removed. The results are shown in the figure below and will be included in the revised manuscript.

[Figure]

The other results are corrected accordingly. The difference between annual cycles of AMET from ERA-Interim and JRA55 becomes smaller, but the low frequency AMET remains different. Hence the study of the difference between the chosen reanalyses remains informative. As the main point of this paper is to raise a caveat about using low frequency MET from reanalyses, the major results in our paper remains the same. Changes are made to the regressions mainly.

*(B) Impacts on the Arctic (section 3.3) The problem of this section is that it gives a few quick examples of regressions of characteristics of the Arctic climate on energy transports, but that they are too short to be really useful. - Show at least the winter and summer seasons, as results could be quite different. - Why show SLP and temperature for AMET, but sea-ice for OMET? Why not show the same variables for both for comparison? (at least sea ice and slp).*

We agree with the reviewer more results are needed to provide more insights about the link between MET and the Arctic and potentially causal relations. Hence, we will include regressions of SLP, SIC for both AMET and OMET in summer and winter and interpret them. In summary they are shown here already and we will further explain these regressions in our revised manuscript:
(the strange-looking colors near the pole for regressions on sea ice in ERA-Interim are due to the quality of sea ice fields in ERA-Interim, see our response to the last small remark.)

[Figure]

*Regression of SLP (a-c), 2 meter temperature (T2M) (d-f) and SIC anomalies on AMET anomalies at 60 N at interannual scale with no time lag in winter (DJF). The monthly mean fields are used here after taking a running mean of 5 years. From left to right, they are the regression of fields on AMET of ERA-Interim, MERRA2 and JRA55. The green contours indicate a significance level of 95%.*

[Figure]

*Regression of SLP (a-c), 2 meter temperature (T2M) (d-f) and SIC anomalies on AMET anomalies at 60 N at interannual scale with no time lag in summer (JJA). The monthly mean fields are used here after taking a running mean of 5 years. From left to right, they are the regression of fields on AMET of ERA-Interim, MERRA2 and JRA55. The green contours indicate a significance level of 95%.*

[Figure]

*Regression of SLP (a-c), 2 meter temperature (T2M) (d-f) and SIC anomalies on OMET anomalies at 60 N at interannual scale with ocean leading by one month in winter (DJF). The monthly mean fields are used here after taking a running mean of 5 years. From left to right, they are the regression of fields on OMET of ORAS4, GLORYS2V3 and SODA3. The green contours indicate a significance level of 95%.*

[Figure]

*Regression of SLP (a-c), 2 meter temperature (T2M) (d-f) and SIC anomalies on OMET anomalies at 60 N at interannual scale with ocean leading by one month in summer (JJA). The monthly mean fields are used here after taking a running mean of 5 years. From left to right, they are the regression of fields on OMET of ORAS4, GLORYS2V3 and SODA3. The green contours indicate a significance level of 95%.*

**Response to the other remarks**

*p.3 line 24: "it is preferable that they incorporate the latest..."*

*p.3 line 25: "they better not resemble each other" well, you certainly hope that different reanalyses would be consistent with each other!*

Thanks for pointing out this. The sentence is very confusing indeed. Here we want to emphasize we try to include various reanalyses data sets to make the comparison more informative. We have deleted this sentence.

*- section 2.3.1 : what is the value of Lv used here ? Lv varies with T in nature, but not necessarily in models... Not sure what's used in reanalyses.*

Same as recent studies (e.g. Mayer et. al., 2017, Trenberth & Fasullo, 2018), we use fixed Lv = 2264.67 KJ/Kg. We include this in the method section now.

Mayer, M., Haimberger, L., Edwards, J. M., & Hyder, P. (2017). Toward consistent diagnostics of the coupled atmosphere and ocean energy budgets. Journal of Climate, 30(22), 9225-9246.

Trenberth, K. E., & Fasullo, J. T. (2018). Applications of an updated atmospheric energetics formulation. Journal of Climate, 31(16), 6263-6279.

*- p.7 : there may also be issues due to different horizontal advection schemes used in the reanalysis and in the post-treatment.*

Thanks for pointing out this. We have updated this part with "All the chosen atmospheric reanalyses use Semi-Lagrangian advection schemes but this is not the case for MERRA2.", just to be accurate.

*- p.8, line 6 : unit should be J/(kgC) or J.C-1.kg-1.*

We will correct it.

*- p.8 : reference temperature. If the unit of potential temperature Theta is in C, then substracting a reference temp. of 0 C does not accomplish anything. Are you instead converting Theta from K to C to avoid cancellation of large terms problems? This is very unclear.*

Reference temperature is used to account for the recirculation and the mass imbalance. Normally people take reference temperature as 0 deg C (e.g. Zheng and Giese, 2009). Since the oceanic reanalyses used here already saved theta in deg C, this actually would not affect the computation. But conceptually a reference temperature should be taken into account.

Zheng, Y. and Giese, B. S.: Ocean heat transport in simple ocean data assimilation: Structure and mechanisms, Journal of Geophysical Research: Oceans, 114, 2009.

*- p.9, l25: The differences in resolution are actually small. There must be another explanation to these variations, which are key (main point A)*

Now we learn that the noise is due to the choice of finite difference scheme. We will delete this.

*- p.10, l10: In ocean models that are not eddy-resolving, there is both an eddy advection (Gent-McWilliams) and a diffusive heat transport, which can be significant compared to the resolved one. How were these incorporated in these analyses? They absolutely need to be taken into account.*

Thanks for the comment. In ORAS4, an eddy parameterization scheme from Gent-Mcwilliams (1990) is implemented. The implementation of this eddy parameterization scheme can lead to a big difference in volume transport and heat transport, compared to eddy-permitted models (Stepanov & Haines, 2014). However, in this case the computation of OMET with ORAS4 doesn't include the contribution from eddy-induced velocity as the fields related to the use of eddy advection schemes are not saved by ORAS4. We will include such information in our revised manuscript.

Stepanov, V., & Haines, K. (2014). Mechanisms for AMOC variability simulated by the NEMO model. Ocean Science, 10(4), 645-656.

*- section 3.2 and the accompanying figures for the atmosphere is a bit pointless given the low quality except for ERA-interim.*

After we updated the results now these comparisons become informative. We will update this part with newly computed fluxes.

*- p11, bottom: Are we looking in this section at the total OMET, or only the Atlantic OMET? It would probably be more interesting to look at Atlantic only at 60 N (in line with section 3.3), although knowing the relative roles could be good. Same question for page 13.*

Thank you very much for the comment. We are looking at the Atlantic OMET in this part.

Below we show the OMET at 60N in the Pacific and Atlantic from ORAS4. It can be noticed that the OMET in the Atlantic is much larger than that in the Pacific. Note that the Atlantic OMET dominates both the mean value and the variability.

Actually, we also tried the same regressions with total OMET. The results are almost the same.

[Figure]

*- p12 : any idea about why ORAS4 seems to work best ?*

Given the big differences between chosen oceanic reanalyses as well as many factors that could potentially influence the results, we cannot explain conclusively why ORAS4 seems to work best here.

*- p13, l28 : why 5-years timescale? Is it specific to the Arctic? Are the regressions on Fig.13 for 5-yr filtered data, or just year-to-year?*

We are focusing on the influence of low frequency signals of OMET on the Arctic. Early studies with climate models also indicate that there are strong correlations between Arctic sea ice and OMET variations at interannual to decadal time scales (Van der Swaluw et al.,2007; Jungclaus and Koenigk, 2010). We take 5-years timescale because reanalysis time scales are relatively short, but we do intend to study variability beyond the annual time scales. Even longer windows for filtering would lead to a too small sample. Figure 13 is the regression with 5 year filtered data. We will add more details to the caption.

Van der Swaluw, E., Drijfhout, S., and Hazeleger, W.: Bjerknes compensation at high northern latitudes: The ocean forcing the atmosphere, Journal of climate, 20, 6023–6032, 2007.

Jungclaus, J. H. and Koenigk, T.: Low-frequency variability of the arctic climate: the role of oceanic and atmospheric heat transport variations, Climate dynamics, 34, 265–279, 2010.

*- Figure 1 :this is hardly commented in the text: seasonal cycle, low contribution of LH (especially to the seasonal cycle)... By the way, I would replace this figure by either the components of the time-mean transport as a function of latitude, or by the mean seasonal cycle at 60 N of the different components. (With figures for the interannual variance if needed)*

Thank you for the comment. We agree that the figure is not quantitatively clear for the seasonal cycles. We will replace it with mean AMET at 60N as a function of month.

*- Figure 2: The dashed-line is built using annual means?*

Yes, the dashed-lines are also annual means. Here we use solid lines and dashed lines to show AMET and OMET, respectively.

*- Figure 3: panel b) needs confidence intervals, based on interannual variance. For panel a), do the standard deviations include the seasonal cycle or are they for interannual anomalies?*

We will include confidence intervals for figure 3b.

In figure 3a the standard deviations include the seasonal cycle. (The standard deviations for interannual anomalies are included in figure 3b)

*- Figure 4: not sure what is the point of these figures, apart from showing that the high-res analyses have eddies? It would be easier to copare maps integrated to the same low resolution, or time-means, and also to see the GM components.*

As these oceanic reanalyses use different curvilinear grid, to integrate them to the same low resolution can introduce errors. We agree that the figures here are a bit pointless. We will delete these figures and only explain these in the text.

*- Figure 6 could easily be replaced by the figures given in the text. Note that latent heat transport may not contribute much to the differences because it's low to begin with, but also because it's concentrated in the near-surface layers, so does not suffer too much from slight mass flux imbalances.*

Thanks for your comment. We will replace these figures by texts.

*- Figure 11, caption: I guess it is "interannual" time-scale? (i.e. year-to-year variability)*

Here we use annual scale to refer to the 1-year filter we applied to the data. By saying interannual scale, we mean a 5-year filter is applied. Sorry for the confusion.

*- Figures 11-13: consistent time-scales and variables, with different seasons would be nicer. Also, the green shading masks the color underneath, making it hard to read. Same-color shading maybe?*

We will include consistent time scales for the regressions of AMET and OMET on SLP and SIC in summer and winter. We will change the green shading to contour lines to make it easy to read.

*- Figure 13b: there are strange-looking colors (opposite the rest of the Arctic...) near the pole in all 3 plots (abrupts changes of sign...) Is this an artefact?*

No, this is not an artefact. There are some quality-issues with sea ice in ERA-Interim, which can be inferred from an evaluation of reanalyses data sets concerning near-surface fields (Lindsay et. al., 2014), as ERA-Interim air-sea flux fields account for local sea ice concentration (IFS DOCUMENTATION – Cy31r1, Chapters 3 and 7). Such strange colors near the pole do not exist in figures with sea ice fields from MERRA2 and JRA55 (see response to major point 2). That's why a 'disc' appears close to the pole in some plots. We will add a note to this issue in the text.

Lindsay, R., Wensnahan, M., Schweiger, A., & Zhang, J. (2014). Evaluation of seven different atmospheric reanalysis products in the Arctic. Journal of Climate, 27(7), 2588-2606.

Again, thank you so much for the comments. We are sincerely gratitude to every point the reviewer made to improve this paper. Thanks a lot!

With best regards

Yang Liu, Ben Moat, Jisk Attema and Wilco Hazeleger

[revised manuscript text omitted]
} (c_p T + L_v q + gz + \frac{1}{2}\mathbf{v}\cdot\mathbf{v})[(1-q)c_p T + L_v q + gz + \frac{1}{2}\mathbf{v}\cdot\mathbf{v}]v\frac{dp}{g}dx \tag{2}$$

with $p_t$ the pressure level at top of the atmosphere ($Pa$) and $p_s$ the pressure at the surface ($Pa$). Since we work on the native hybrid model coordinate with each atmosphere reanalyses product, the equation can be adjusted as follows (see Graversen (2006)):

$$E = \oint_{\Phi=\Phi_i} \frac{1}{g} \int_{0}^{1} (c_p T + L_v q + gz + \frac{1}{2}\mathbf{v}\cdot\mathbf{v})[(1-q)c_p T + L_v q + 
[revised manuscript text omitted]

Monthly mean of OMET in January 1996 from three ocean reanalyses products (a) ORAS4 (b) GLORYS2V3 and (c) SODA3.

[Figure]

[Figure]

**Figure 4.** Time series of zonal integral of OMET at $60°$N without/with low pass filter. (a) The original time series (top) and (b) the ones with low pass filter (bottom) include signals from ORAS4 (blue), GLORYS2v3 (red)and , SODA3 (green) and the OGCM hindcast (yellow). For the low pass filtered ones, we take a running mean of 5 years. The shades represent the confidence intervals with one standard deviation. $\sigma$ is the standard deviation and $\mu$ is the mean of the entire time series.

Difference in total AMET and each component between ERA-Interim, MERRA2 and JRA55 at 60°N in the same period. (a) The deviation between ERA-Interim and MERRA2, as well as (b) the deviation between ERA-Interim and JRA55, are defined as the component-wise subtraction. The unit is Peta Watt (PW).

[Figure]

**Figure 5.** Difference in temperature, meridional wind velocity and temperature transport between MERRA2 and ERA-Interim at 60°N. The vertical profile of (a) temperature difference and (b) meridional wind velocity difference are calculated from the climatology of each fields from 1994 to 1998, respectively.

[Figure]

**Figure 6.** OMET estimated from ORAS4 (blue), GLORYS2V3 (red), SODA3 (green) and the  OGCM hindcast (orange) compared to the RAPID ARRAY observation (gray) at 26.5°N across the Atlantic basin. The time series of OMET is presented in (a). The statistical properties are shown in (b) Taylor Diagram, including bias, correlation (blue), standard deviation (black) and root mean square deviation (green). $\sigma$ is the standard deviation and $\mu$ is the mean of the entire time series.

[Figure]

**Figure 7.** OMET estimated from ORAS4 (blue), SODA3 (green) and compared to the OSNAP observation (gray) at subpolar Atlantic basin. The range of uncertainty from OSNAP observation is marked by the red shade. $\sigma$ is the standard deviation and $\mu$ is the mean of the entire time series.

[Figure]

[Figure]

**Figure 8.** Time series of (a) ocean heat content (OHC) and (b) OHC anomalies with a low pass filter at the polar cap. The OHC is  integrated from surface to the bottom between 60°N and 90°N. It is estimated from ORAS4 (blue), GLORYS2V3 (red), SODA3 (green) and the  OGCM hindcast (yellow). The shades represent the confidence intervals with one standard deviation. $\sigma$ is the standard deviation and $\mu$ is the mean of the entire time series.

[Figure]

**Figure 9.**  Regressions of sea level pressure(SLP), 2 meter temperature and sea ice concentration anomalies on AMET anomalies at $60°$N in winter (DJF) at  interannual time scales with no time lag.  The monthly mean fields are used here after taking a running mean of  5 year. Both the 2 meter temperature and sea ice concentration are detrended. From left to right, they are the  regressions on AMET of (a, d, g) ERA-Interim, (b, e, h) MERRA2 and (c, f, i) JRA55. The green  contour lines indicate a significance level of 95%.

[Figure]

**Figure 10.**  Regressions of sea level pressure, 2 meter temperature  and sea ice concentration anomalies on  OMET anomalies at 60°N in winter (DJF) at interannual  time scales. OMET leads the fields by one month. The  2 meter temperature, sea ice concentration and OMET are detrended. From left to right, they are the  regressions on  OMET of (a, d, g) ORAS4, (b, e, h)  GLORYS2V3 and (c, f, i) SODA3. The green  contour lines indicate a significance level of 95%.

[Figure]

**Figure 11.**  Regressions of sea level pressure, 2 meter temperature and sea ice concentration  anomalies on  AMET anomalies at 60°N  in summer (JJA) at interannual time scales with no time lag. The monthly mean fields are used here after taking a running mean of 5 year. Both the  2 meter temperature and  sea ice concentration are detrended. From left to right, they are the  regressions on  AMET of (a, d, g) ERA-Interim, (b, e, h)  MERRA2 and (c, f, i) JRA55. The green  contour lines indicate a significance level of 95%.

[Figure]

**Figure 12.** Regressions of sea level pressure, 2 meter temperature and sea ice concentration anomalies on OMET anomalies at 60°N in summer (JJA) at interannual time scales. OMET leads the fields by one month. The 2 meter temperature, sea ice concentration and OMET are detrended. From left to right, they are the regression on OMET of (a, d, g) ORAS4, (b, e, h) GLORYS2V3 and (c, f, i) SODA3. The green contour lines indicate a significance level of 95%.

---

## Referee Report (RR1)

Review for "Synthesis and evaluation of historical meridional heat transport
from midlatitudes toward the Arctic" by Liu et al.

The paper compares the meridional energy transports by the atmosphere and ocean
at high latitudes in different datasets, with the general aim of validating
their use in the analysis of observed slow (decadal ?) variability.

Compared to the first version, some major problems in the computation of the
transports have been corrected. There is however a number of smaller issues
remaining, so I would recommend at least a minor revision.

A list of specific comments follows, but more generally the paper needs to be
edited to improve the English : it is barely understandable in places. Examples
are page 11, lines 7-10, or page 10, lines 26-29.

- ERA-I and JRA-55 now seem very close to each other in most respects (mean,
  variations, regressions with surface variables...). This should be
  highlighted in the conclusion / abstract ?
  Merra 2 behaves very differently, and seems still noisy : a problem remaining
  in the computation ?

- page 6, line 24 : the moist static energy is H + I + gz, not H.

- page 7, l9 (advection schemes) : this should come later (as a potential
  source of errors)

- page 9, line 7 : the sentence should be "note that we have access to
  sub-monthly data only for SODA3. The computation of OMET in GLORYS using
  monthly data could miss part... eddies, while ORAS4 does not have explicit
  eddies (insert missing GM comment here). There is no causality between having
  data for SODA3 and the smoothing in GLORYS.

- Figure 1/2 : it would be useful to see a decomposition into components (at
  least dry static / latent) as a function of latitude.

- fig 3b : ERA-I and JRA55 look similar to me (at least compared to MERRA). The
  text should reflect thet.

- page 10, lines 10-20 : it would be useful to have a figure of OMET as a
  function of latitude, to support the discussion in this paragraph. Also,
  presumably the GM transport in ORAS4 would compensate ?

- page 10, line 15 : eddy-induced velocity contributes to the heat transport,
  not to the volume transport. (works like en overturning streamfunction).

- page 11, lines 20-30, and figure 5 : This discussion assumes that the
  differences in AMET are due to differences in the climatological-mean v or T
  (not knowing which), but why would this be true when the total transport is
  dominated by transients ? This figure seems unnecessary as not much can be
  concluded from it.

- page 12, line 30- (OHC) : why would polar cap OHC be a sign of Arctic
  Amplification (not just Arctic warming)? Note that the observed increasing
  trend of OHC can also be due to surface heat fluxes, indeed there is a
  downward trend of OMET at 60° over the same period...

- Section 3 : the use of "interannual" is usually year-to-year variability. To
  use it for a 5-yr smoothing (intending ~ decadal signals) is a bit
  misleading.

- section 3 (bis) : a striking point in these results is the similarity between
  ERA-I and JRA, and ORAS4/SODA3. For the latter, it would be good to know if
  it could be due to similar surface fluxes used, or to the model / data
  assimilation.

- page 14, lines 10-15 : Why "an increase in OMET is related to warm and humid
  air transport over the North Atlantic" ??  My impression on figure 10 is that
  an increase in OMET leads to sea-ice melt and increase in T2m around the
  Nordic seas. In addition, there is an AO/NAO-like SLP anomaly (that can be
  cause or consequence) with the associated large-scale temperature pattern
  (North AM-Greeland / Siberia dipole).

- figure 11 : this seems broadly consistent with a colder Arctic : colder temp,

more ice, high pressure. Is this causing the increase in AMET?

– figure 12 : In the sea ice regressions, it looks like there are sea ice
  trends in areas with no ice in summer... Also, values are very large, may be
  a scale of % per 0.01 PW would be more adapted ?

---

## Author Response (AR2)

Dear Editor Prof. Dr. Lohmann,

Many thanks for the time and the effort you put on processing and reviewing our paper. We have revised our manuscript based on the remarks from two anonymous reviewers. The main changes are to add more details based on the major comments in the manuscript and to edit the whole manuscript for language.

Again, we would like to thank you for reviewing our work and contacting reviewers. We look forward to hearing from you about the progress of the following review procedure. We are sincerely gratitude to everything you made to improve this paper.

With best regards

Yang Liu, Jisk Attema, Ben Moat, and Wilco Hazeleger

Response to reviewer #1,

We would like to thank the reviewer for reviewing our manuscript and providing many constructive comments. Similar as in previous review, we address the comments point-by-point and our response is given below. The main changes are to add more details based on the major comments in the manuscript and to edit the whole manuscript for language.

(the *original comments* are given by *Italic gray text*, and each follows our response in plain text. The page and line numbers of changes made to the text are listed below each response.)

**Response to the major comments**

*In the conclusions, the authors recommend not using reanalyses for energy transports diagnostics. I think this statement is too general. While I agree that data assimilation systems used in reanalyses do not explicitly conserve energy, energy budget diagnostics still are a very useful method to assess the quality of the products. And indeed, some diagnostics show quite good agreement, e.g. the mean annual cycle of the atmospheric energy transports, as shown by the authors. Moreover, other studies found quite good agreement also for oceanic transports (Uotila et al. 2019; Mayer et al. 2019) or seasonal trends in the budgets (Mayer et al. 2016). For the decadal variability, it should be kept in mind that relative changes on these timescales are really small (a few percent of the total transport), which is very hard to get right given the permanently evolving observational system (in both atmosphere and ocean). So I suggest to soften the conclusions (and abstract) in that regard and say that care should be taken when doing this kind of evaluations and that robustness of results must be assessed through intercomparison (as the authors did).*

We thank the reviewer for this comment and call for a more nuanced statement. We were not implying that these reanalyses were not recommended for evaluation of energy transports. We only would like to put a caveat on the low frequency signals of energy transport derived from the reanalysis data. But we do agree with the reviewer that this should be made clear in the texts.

We now updated both the abstract and the conclusion with a recommendation for using reanalyses to calculate energy transport and evaluate products, and an alert on energy transport diagnostics at large (e.g. decadal) time scales and the robustness of results.

Page 1, lines 17-22 and page 19, lines 4-9

*P6L15: Attributing differences between these very different ocean products to resolution will be difficult as there are many other differences, such as in the forcing and data assimilation.*

We thank the reviewer for the comment. We agree that the NEMO ORCA hindcast is different in many aspects. We didn't aim to emphasize the difference in resolution. Therefore, in this paragraph more details are provided about the NEMO ORCA hindcast. To avoid confusion, we reformulate it without saying "high resolution".

Page 6, lines 27

*Section 2.3.1: The authors nicely discuss mass imbalances and how they can be corrected for. However, the authors do not describe which fields they use in practice. While earlier studies have detailed this for ERA-Interim (and JRA55 is very similar structure-wise), I am not aware of a detailed description of the*

*mass adjustment for MERRA2. This would certainly be of interest, given that MERRA2 is very different from the ECMWF reanalyses in many regards. Also, suboptimal implementation of the correction for MERRA2 may also explain why ERA-Interim and JRA55 results are quite similar, while MERRA2 results look different.*

We would like to thank the reviewer for the comment. The fields used in practice are shown in equation (4) and (5), which are surface pressure, meridional and zonal winds, and specific humidity. We will stress this in the text.

In terms of the discretization and grid incorporated by the dynamical core, MERRA2 is very different from ERA-Interim and JRA55. The dynamical core for MERRA2 is the GEOS-5 model and it computes all fields on a cubed-sphere grid with an resolution of 50kmx50km (see their official file specification: https://gmao.gsfc.nasa.gov/pubs/docs/Bosilovich785.pdf). However, the data collections are saved only on the latitude-longitude grid after interpolation (source of data via GES DISC: https://disc.gsfc.nasa.gov/datasets/M2T3NVASM_5.12.4/summary?keywords=MERRA-2). Because of the interpolation, the data cannot be transferred back to the cubed-sphere grid without loss of information. Besides, vector field computations on the cubed-sphere grid are not divergence free due to the implementation of finite volume discretization methods (Putman and Lin, 2007). Therefore, we transferred MERRA2 fields to the spectral domain and performed vector field computations via spherical harmonics to minimize the loss. This might explain the difference between the results from MERRA2, ERA-Interim and JRA55. However, due to the difference in resolutions, dynamical cores and data assimilation methods, a causality cannot be determined.

Even if we would have MERRA2 data on cubed-sphere grid, there is no proper tool to perform the computation of divergence, gradient and inverse Laplacian, which are required by mass budget correction.

We added this discussion to our paper.

Page 9, lines 3-11

Reference

Putman, W. M., & Lin, S. J. (2007). Finite-volume transport on various cubed-sphere grids. Journal of Computational Physics, 227(1), 55-78.

*P9L9: the varying sea surface height is not a strong argument against doing a barotropic mass adjustment in the ocean: most of the local sea level change is due to steric changes, rather than ocean mass changes. So ocean bottom pressure would be the appropriate quantity. Also, as I wrote in my first review, we tested these things in our own work and I can say that mass imbalances are not present in the NEMO model, i.e. the divergence of ocean currents is consistent with bottom pressure changes up to very high accuracy (see also the NEMO documentation). What you could say instead is that there is a mass imbalance (in the sense of non-vanishing lateral divergence) in the stemming from P-E and small hard-to-diagnose budget terms.*

We thank the reviewer for the suggestion. We now explain it with P-E and small, but hard to diagnose budget terms.

Page 9, lines 26-27

*P9L21: It is unclear to me how you do the calculation. You are writing about "integrals along the zonal direction of the native grid". Instead, one has to perform "zig-zag" line integrals (including zonal and meridional transports to close the line) on the native grid in order to get transports across a circle of latitude. Can you clarify how you do these computations?*

Sorry for the unclear description. Indeed, we followed a zig-zag setup to take the zonal integrals on curvilinear grid. The method is explained by Outten et al., (2018) in their Figure 2. We updated the text and include this reference.

Page 10, lines 8-10

Reference

Outten, S., I. Esau, and O. H. Ottera°, 2018: Bjerknes compensation in the cmip5 climate models. Journal of Climate, 31 (21), 8745–8760.

*P9L23: Use of monthly ocean data for computation of transports is particularly problematic in regions with eddy activity. The authors could use SODA3 data (from which they do have daily data) to estimate the error of neglecting sub-monthly variability as a function of latitude.*

We would like to thank the reviewer for the comment. We agree with the reviewer that monthly data is problematic for eddies. In this case, ORAS4 and GLORYS2V3 are monthly data, while SODA3 is 5 daily averaged. This combination already provides some insight into the contribution from eddy transports. It would be nice to have either daily data or monthly data with SODA3 to gain more knowledge about the contribution from eddies. However, for SODA3, only 5 daily data is available on model (original) grid. Daily data is not available. Monthly data has been interpolated to latitude-longitude grid, thus it cannot be used for comparison. (see the link here for the access to SODA3 data with surface forcing coming from ERA-Interim https://www.atmos.umd.edu/~ocean/index_files/soda3.4.1_mn_download.htm). Therefore, we cannot check the sub-monthly variability of energy transport within the same reanalysis dataset.

*Section 3.3: Many results in this section obviously are not robust at all. While it is of value to point this out, I do not think it makes sense to discuss them too much in detail. There is too much discussion of things that may be spurious. Also, I am unsure whether it is meaningful to regress pan-arctic atmospheric fields onto OMET, where OMET leads by one month. How would OMET impact SLP one month later? It may make more sense to, e.g., regress OMET with SLP at zero lag, as surface winds drive ocean surface currents. The reverse impact of OMET on SLP is probably much weaker.*

We thank the reviewer for the comment. The extended analysis in section 3.3 is requested by reviewer #2. It helps to investigate the physical plausibility by comparing the relation between MET and the Arctic within each reanalysis product. But we do agree with the reviewer that most of the results are not robust. Therefore we emphasized that we aim to study physical plausibility of fields associated with OMET variations. Indeed, numerical model studies indicate such relationships.

We now move the summer regressions to the supplementary material and reduce the length of the discussion.

Regarding the reviewers' questions about regressions of SLP on OMET, those are not entirely about the wind-driven effect. It seems that the ocean could affect atmospheric fields thermodynamically via OMET

convergence -> net surface flux -> SLP (no causality proved), which could explain the time lag. We noticed that the regression coefficients peak when the ocean leads by one month, for both the regressions of net surface fluxes (see figures below) and SLP on OMET.

Section 3.3 from page 14-17

[Figure]

*Regression of net surface fluxes on OMET at 60N at interannual scale in winter with ocean leading by 1 month.*

*I also have to come back to the use of sea ice data. I would at least recommend to perform all computations against an "independent" satellite products, to check how robust the results are. It is not necessary to show extra figures, but it should be checked.*

We thank the reviewer for the comment. We performed the same regressions with AMET and OMET at 60N on NOAA/NSIDC Climate Data Record of Passive Microwave Sea Ice Concentration (Version 3, https://nsidc.org/data/G02202/versions/3). The results are very similar to our results with sea ice from reanalyses. For consistency, the regressions on AMET are shown below.

***From left to right (ERA-Interim, MERRA2 and JRA55)***

[Figure]

*Regression of NOAA/NSIDC Climate Data Record of Passive Microwave Sea Ice Concentration on AMET in winter (DJF).*

[Figure]

*Regression of NOAA/NSIDC Climate Data Record of Passive Microwave Sea Ice Concentration on AMET in summer (JJA).*

**Response to the minor comments**

*P1L6: 2010 is inconsistent with Fig 4b and P11L32, which indicates good agreement from ~2007 onward*

Corrected.

Page 1, line 6

*P1L11: "among all the chosen" sounds a bit bold given that you only use 3 atmospheric reanalyses. Better to remove "all".*

Deleted.

Page 1, line 9-10

*P2L27: remove "of" before "OHC"*

Corrected.

Page 3, line 5

*P2L30: insert "the" before "understanding"*

Corrected.

Page 3, line 9

*P3L8: Why "ARRAY" in capital letters? It is not an acronym, so I would recommend "array".*

Corrected.

*P4L3: "higher" (than what?) -> "high"*

It is "high" here.

Page 4, line 13

*P4L3: I disagree with "due to the need". Budget diagnostics are certainly not the main reason for having reanalyses with high resolution. In fact it is the other way round: they have high resolution and thus "enable" energy budget diagnostics*

Sorry for the confusion. Here we mean "due to our needs for energy budget diagnostics". We now reformulate it.

Page 4, line 13-14

*P4L4: "It is preferable…" This statement is obvious and can be removed.*

Deleted.

*P4L16: "generates data using 4D-Var assimilation" -> "generates atmospheric state estimates using 4D-Var data assimilation"*

Updated.

Page 4, line 26

*P4L20: Could add that this data comes on a 256x512 gaussian grid*

We added it.

Page 5, line 1

*P4L26: add "Incremental Analysis Update (IAU)" before "assimilation"*

We added it.

Page 6, line 8

*P4L27: preceding -> predecessor*

We think the reviewer mean Line 17. Corrected.

Page 4, line 27

*P5L2: insert „the" before „Japan"*

Corrected.

Page 5, line 13

*P5L4: "assimilated observations" -> "assimilated upper air observations"*

It is "assimilated upper air observations" here.

Page 5, lines 15-16

*P5L5: "level" -> "grid"*

Corrected.

Page 5, line 17

*P5L15: "at" -> "in the"*

Corrected.

Page 5, line 27

*P5L16: The last sentence contradicts the statement above that high temporal resolution is needed for the presented diagnostics*

ORAS4 has only monthly resolution but it is good in many other aspects. We think our description is ok.

*P5L21: approximate -> approximately*

Corrected.

Page 5, line 2

*P5L27: What does "mainly" mean here?*

Because part of their work was done in TAMU, and many other joint organizations (e.g. NOAA/GFDL, NOAA/NESDIS, etc.). See https://www.atmos.umd.edu/~ocean/

*P6L9: remove "of" after "comprises"*

Corrected. Sorry for the typo.

Page 6, line 21

*P6L18: remove "climatological"*

This is the description from paper Moat et al., (2016). It is better to keep it.

*P9L6: maybe better "difference" instead of "residual"*

Corrected.

Page 9, line 23

*P9L14: I presume this is because recirculation is cancelled out?*

Agree, quite likely to be, but not proved.

*P9L26: insert "to" before "compare"*

Corrected

Page 10, line 16

*P10L1: delete "simply"*

Deleted.

Page 10, line 20.

*P10L7: Is this just a convoluted way of saying that autocorrelation is taken into account?*

We changed it.

Page 10, line 20-25

*P10L8: I suggest adding "relatively" before short*

Corrected.

Page 10, line 23

*P10L15: What is meant by "from 1 to 5"? Is it 1 or 5 or something in between?*

Sorry for the confusion. It is exactly 5 years running after revision. We corrected it.

Page 10, line 30

*P10L19: This is still inaccurate. There is a transport from regions with positive net TOA radiation to regions with negative net TOA radiation.*

Sorry for the confusion. It is corrected now.

Page 11, lines 3-4

*P11L4: ERA-Interim and JRA55 agree quite well, actually. How about the correlation coefficient?*

The correlation coefficient is 0.82 (while between ERA-Interim and MERRA2 is -0.53). We will add it to the text.

Page 11, lines 18-21

*P11L13: Shouldn't the ocean currents be implicitly affected by this?*

Yes, they should. This is requested by Reviewer #2.

*P11L20: This is just a suspicion. How can you be sure about this statement?*

We only know that the other two models are eddy-permitting with high spatial resolution. But we don't know if they can produce realistic patterns. We agree with the reviewer and we reformulated the sentence "….which might represent more realistic patterns….".

Page 12, lines 1-2

*P12L6: I think you mean "correlation"*

Yes, we do. We reformulate it to make it clear.

Page 12, lines 24-27

*P12L7: please provide the value*

It is 0.21 and the value is given there.

Page 12, line 27

*P12L11: neglectable -> small*

Corrected.

Page 12, line 29

*P12L11: over -> of*

Corrected.

Page 12, line 30

*P12L11: rest of the paragraph is hard to understand, e.g. what is meant by "This is generally the case"?*

Sorry for the confusion. We add more details this paragraph.

Page 12, lines 30-35 and page 13, lines 1-3

*P13L4: "make a priori judge" -> cannot judge a-priori*

Corrected.

Page 13, line 24

*P13L16: NEMO -> OGCM*

Corrected.

Page 14, line 1

*P13L34: can you provide these values also as a column-average temperature?*

We thank the reviewer for the comment. The column-integrated OHC already implies column-averaged temperature. Since most of the oceanography literatures show OHC in Joule, for consistency, we think it suffices to just show OHC here.

*P14L37 vs P14L5: aren't the statements about agreement among the products contradictory?*

Sorry for the confusion. This is an editing error. We revised it again.

Page 14, lines 25-29

*P23L35: I think you mean issue 2 of the ocean state report, which appeared in 2018.*

We thank the reviewer for the comment. It is corrected.

Page 24, line 33

*Figs 9-12: Maybe better to stipple statistically significant values. Also, have fields like SIC been de-trended before performing regression?*

We thank the reviewer for the suggestion. We tried stippling but it masks the color underneath. So, we switched to contour lines. All the fields have been detrended before performing regression, as mentioned in the beginning of section 3.3.

**Response to the general comments**

*I would strongly recommend to let the manuscript check for language. There are several sentences that sound very sloppy and/or are hard to understand (e.g. P2L12, P10L10). I am not an English native speaker – so I will refrain from making suggestions.*

We went over the manuscript very carefully and now the paper has been edited with great care (One of our authors is a native speaker).

*There still seems to be some confusion about the spelling of reanalysis/reanalyses. In my opinion, "reanalyses" should only be used when standing alone and when plural is meant. In combination with*

*"dataset", it should always be "reanalysis", i.e. "reanalysis data set" for singular and "reanalysis data sets" for plural. Please modify accordingly.*

Thanks for the explanation. We correct it.

Again, we would like to express our gratitude to the time and effort that the reviewer spent on our manuscript. We feel the manuscript has improved again!

With best regards

Yang Liu, Jisk Attema, Ben Moat, and Wilco Hazeleger

Response to reviewer #2,

We would like to thank the reviewer for reviewing our manuscript again. Similar as in previous review round, we address the comments point-by-point. The main changes are to add more details based on the major comments and to edit the whole manuscript for language. Our responses are given below:

(the *original comments* are given by *Italic gray text*, and each follows our response in plain text. The page and line numbers of changes made to the text are listed below each response.)

**Response to minor comments**

*A list of specific comments follows, but more generally the paper needs to be edited to improve the English: it is barely understandable in places. Examples are page 11, lines 7–10, or page 10, lines 26–29.*

We thank the reviewer for the check. We edited the manuscript and reformulated the sentences which would cause confusion.

*– ERA–I and JRA–55 now seem very close to each other in most respects (mean, variations, regressions with surface variables...). This should be highlighted in the conclusion / abstract? Merra 2 behaves very differently, and seems still noisy: a problem remaining in the computation?*

We thank the reviewer for this good question. In terms of the discretization and grid incorporated by the dynamical core, MERRA2 is very different from ERA-Interim and JRA55. The dynamical core for MERRA2 is GEOS-5 model and it computes all fields on a cubed-sphere grid with an resolution of 50kmx50km (see their official file specification: https://gmao.gsfc.nasa.gov/pubs/docs/Bosilovich785.pdf). However, the data collections are saved only on the latitude-longitude grid after interpolation (source of data via GES DISC: https://disc.gsfc.nasa.gov/datasets/M2T3NVASM_5.12.4/summary?keywords=MERRA-2). Because of the interpolation, the data cannot be transferred back to the cubed-sphere grid without loss of information. Besides, vector field computations on the cubed-sphere grid are not divergence free due to the implementation of finite volume discretization methods (Putman and Lin, 2007). Therefore, we transferred MERRA2 fields to spectral domain and performed vector field computations via spherical harmonics to minimize the loss. This might explain the difference between the results from MERRA2, ERA-Interim and JRA55. However, due to the difference in resolutions, dynamical cores and data assimilation methods, a causality cannot be determined.

Even if we would have used MERRA2 data on cubed-sphere grid, there is no proper tool to perform the computation of divergence, gradient and inverse Laplacian, which are required by mass budget correction.

We now add this explanation to our paper and highlight the similarity between ERA-I and JRA55, and the difference with MERRA2 in the abstract and conclusion.

Page 9, lines 3-11

*total transport is dominated by transients ? This figure seems unnecessary as not much can be concluded from it.*

We thank the reviewer for the comment. The idea originates from Figure 2, which illustrates the difference of annual mean AMET as a function of latitude between ERA-Interim and MERRA2. We would like to understand the source of the differences. We agree with the reviewer that at mid-latitudes the total energy transport is mainly eddy driven. However, given the differences in annual mean AMET, it is instructive to show the difference in mean v & T fields. So, we think this figure is informative.

*– page 12, line 30– (OHC): why would polar cap OHC be a sign of Arctic Amplification (not just Arctic warming)? Note that the observed increasing trend of OHC can also be due to surface heat fluxes, indeed there is a downward trend of OMET at 60° over the same period...*

We thank the reviewer for the comment. Increases in surface temperature and OHC are often taken as a sign of AA in many peer reviewed papers (e.g. Serreze and Barry 2011). But we do agree with the reviewer that it might be just Arctic warming and not necessarily a higher warming rate than the global mean temperature change. Moreover, we do not aim to identify a causality here. Therefore, we reformulated it "…could be taken as a sign of AA……".

The downward trend of OMET at 60N is likely related to the AMOC or the energy compensation between the atmosphere and ocean or a combination thereof (Smeed et al., 2014; McCarthy et al., 2015; Oltmanns et al., 2018).

Page 14, lines 20-24

*– Section 3: the use of "interannual" is usually year–to–year variability. To use it for a 5–yr smoothing (intending ~ decadal signals) is a bit misleading.*

We thank the reviewer for the comment. We agree that 'interannual' is a bit unclear. Thus, we will put a note at the beginning of section 3 "…….with a low pass filter of 5 years, which is now referred to as interannual time scales for the rest of the paper".

Page 10, line 30.

*– section 3 (bis): a striking point in these results is the similarity between ERA–I and JRA, and ORAS4/SODA3. For the latter, it would be good to know if it could be due to similar surface fluxes used, or to the model / data assimilation.*

We thank the reviewer for the comment. Actually, for these reanalysis products, the dynamical core, data assimilation method and the data assimilated are all very different and it is very difficult to disentangle them and find causal relations. We have underlined their dynamical core, data assimilations method and surface forcing in the section 2 "Data and methodology". It was emphasized in the text that the chosen oceanic reanalyses systems all use surface fluxes from ERA-Interim and ERA-40.

Section 2.1, pages 4-6

*– page 14, lines 10–15 : Why "an increase in OMET is related to warm and humid air transport over the North Atlantic" ?? My impression on figure 10 is that an increase in OMET leads to sea–ice melt and increase in T2m around the Nordic seas. In addition, there is an AO/NAO–like SLP anomaly (that can be cause or consequence) with the associated large–scale temperature pattern (North AM–Greeland / Siberia dipole).*

We thank the reviewer for the comment. Since a causal relationship was not identified, we cannot put a solid conclusion here, thus only one suggested mechanism similar to this one. We do agree with the reviewer that it is likely to be an increase in OMET and OMET convergence that leads to sea–ice melt and increase in T2m around the Nordic seas, and there is an AO/NAO–like SLP anomaly with the associated large–scale temperature pattern.

We now reformulated it carefully and only describe the patterns following the reviewer's suggestion.

Page 16, lines 7-16

*– figure 11 : this seems broadly consistent with a colder Arctic : colder temp, more ice, high pressure. Is this causing the increase in AMET?*

This could be the case. Since identifying causal relationships is not the aim of this analysis, we cannot say this for sure.

*– figure 12 : In the sea ice regressions, it looks like there are sea ice trends in areas with no ice in summer... Also, values are very large, may be a scale of % per 0.01 PW would be more adapted?*

We checked the regressions of sea ice, especially for the marginal sea ice covered area which could be ice free in summer. Most of the ice free regions in recent decades have sea ice in early 1990s. As the

anomalies were taken by removing the climatology and detrended, the sea ice concentration in these regions are not strictly zero.

We rescaled it by per 0.1PW. Now they look very reasonable.

Figure 10 and S2

Again, we would like to express our gratitude to the time and effort that the reviewer spent on our manuscript. We thank the reviewer for contributing to the improvement of our paper!

With best regards

[revised manuscript text omitted]

---

## Author Response (AR3)

Dear Editor Prof. Dr. Lohmann,

We would like to thank you for the time and the effort you put on processing and reviewing our paper. We have revised our manuscript based on the final remarks from the anonymous reviewer. The main changes are to use the proper latent heat of vaporization ($Lv$ = 2500 KJ/Kg at 0°C) and update all the results related to latent heat transport with the new constant.

Again, we would like to thank you for reviewing our work and contacting reviewers. We look forward to hearing from you about the progress of the following procedure.

With best regards

Yang Liu, Jisk Attema, Ben Moat, and Wilco Hazeleger

Response to reviewer #1,

We would like to thank the reviewer for reviewing our manuscript and providing insightful comments again. The same as last time, we have tried to address your comments point-by-point and our response is given below. The main changes are to use the proper latent heat of vaporization and update all the results related to latent heat transport with the new constant.

(the *original comments* are given by *Italic gray text*, and each follows our response in plain text.)

**Response to the major comments**

*The manuscript has substantially improved compared to earlier versions. There are still a few minor issues that I recommend to address before the paper can be accepted for publication. The only major comment is that the authors still make a very uncommon choice for latent heat of vaporization (namely that at 100°C), which is more than 10% lower than more standard values. This must be corrected, possibly including recomputation of the results if this wrong value was really used and not only stated in the text.*

After a careful check, we agree with the reviewer that we can take a better choice for latent heat of vaporization. Now we use Lv = 2500 KJ/Kg (0°C) instead of Lv = 2264.67 KJ/Kg (100°C) and recompute the latent heat transport with the new value of Lv. Fortunately, the contribution from latent heat transport to the total energy transport is much smaller than the temperature transport and thus the major results are not changed that much. For instance, the difference of 10% changes of latent heat is not noticeable in the Figure 1, which shows the monthly mean meridional energy transport in ERA-Interim (see magenta line with Lv = 2500 KJ/Kg and black line with Lv = 2264.67 KJ/Kg). **We update our results for the whole paper based on the new latent heat transport (still, they are not very noticeable).**

[Figure]

*Figure 1. Monthly mean meridional energy transport in ERA-Interim.*

**Lines: Page 1, 16-21 and page 18, 25-30**

**Response to the minor comments**

*P2L31: There is an additional, more recent reference for this statement: Mayer, M., Tietsche, S., Haimberger, L., Tsubouchi, T., Mayer, J., & Zuo, H. (2019). An improved estimate of the coupled Arctic energy budget. Journal of Climate, 32(22), 7915-7934.*

We added it to our reference. We thank the reviewer for the paper.

**Page 2 Lines 30-31**

*P3L1: "point" --> "point to"*

Corrected.

**Page 3 Line 1**

*P3L32: remove "as a result": your choice is not a result of the statements before.*

Corrected.

**Page 3 Line 32**

*P5L3: "old" is relative. One could view ORAS4 as "old" as well, given ORAS5 is available already. Maybe better say "predecessor"*

We thank the reviewer for the comment. We re-write the sentence with "predecessor".

**Page 5 Line 2**

*P5L6: the forcing changed to operational forcing in 2010, see Balmaseda et al. (2013)*

We thank the reviewer for the comment. We update this part with the information provided by the reviewer.

**Page 5 Lines 6-7**

*P6L14: remove "the" before "produced"*

Corrected.

**Page 6 Line 15**

*P7L13: I understand now the reason for the confusion. The authors provide Lv at the boiling point of water, where Lv is significantly lower than at lower temperatures: see graph at https://www.engineeringtoolbox.com/water-properties-d_1573.html*

*This is a very unusual choice. Usually, Lv at 0°C is used as this is a much more typical air temperature. See, for example, the description of ECMWF's IFS part IV: physical processes. See chapter 12: the choice in the IFS is 2.508e6J/kg.*

We thank the reviewer for the comment. Sorry for the confusion. Now we take Lv at 0°C and recompute all the latent heat transport in this paper. See our explanation for the major point.

**Page 7 Line 9**

*P9L14: add °C to "0"*

Corrected.

**Page 9 Line 14**

*P10L25: change sentence to something like "One possible explanation…"*

We thank the reviewer for the comment and we reformulate this sentence.

**Page 10 Line 25**

*P11L11: please state here whether you did check surface fluxes and OHC changes*

We thank the reviewer for the reminder. We did check the OHC changes and actually we found the large differences in OHC among chosen products. We updated this in the text.

**Page 11 Lines 11-12**

*P11L15: "source" --> "sources"*

Corrected.

**Page 11 Line 16**

*P11L19: remove "With linear regression,"*

Corrected.

**Page 11 Line 20**

*P12L26: The fact that OGCM does not assimilate ocean data and still agrees well with observations suggests that the surface forcing is a very important driver of OMET variability. You should explicitly state this.*

We thank the reviewer for the comment and we explicitly stated this now.

**Page 11 Line 20**

*P12L26: The sentence "To conclude, the heat transport at 26:5N is too low in these products" is right, but you could add a statement that two products appear to have reasonable variability.*

We agree with the reviewer for the comment and we added it to the text now.

**Page 12 Lines 26-29**

*P12L29: I suggest to remove "quite"*

Corrected.

**Page 12 Line 32**

*P13L6: How can the OHC of OGCM be almost twice as high as those of the other products? This would suggest twice as high column-average temperatures in OGCM. This cannot be right. There must be an inconsistency in the choice of integration depth or land-sea masking. Please check and correct.*

We thank the reviewer for the notice. We checked our computations of OHC with OGCM and found nothing abnormal. We also checked other strong signals, like AMOC or heat transport, and they show good agreement with reanalyses. We notice that concerning the OHC between 60N and 70N, the OGCM hindcast and ORAS4 compare well but the other two reanalyses are quite different (see Figure 2 below). While in Figure 8a in our paper the OGCM hindcast does seem to be the outlier, considering the OHC between 60N and 90N. This indicates that the difference comes from the Arctic (between 70N and 90N) in the OGCM hindcast. It might be associated with changes to the sea ice distribution (see Moat et. al., 2016). It's not obvious that the model is wrong since there aren't so many observations in the Arctic to constrain the reanalyses, and they may make different assumptions about sea ice and are also low resolution compared to the OGCM hindcast. Either way, it seems a bit more complex than just claiming the OGCM hindcast is bad compared to the reanalyses. We added a brief explanation in the text.

[Figure]

*Figure 2. OHC between 60N and 70N.*

**Page 13 Lines 9-13**

*P16L26: add "interannual variability of" before "AMET"*

Added.

**Page 17 Line 5**

*P16L32: "large" --> "long"*

Corrected.

**Page 17 Line 11**

*P16L32-33: You find that the annual cycle of the products agrees well, but interannual (five-yearly filtered) variability not. What is the maximum time scale that can be viewed as robust? How about annual means anomalies?*

We thank the reviewer for this good question. We checked the annual means anomalies as well, but they do not agree among these reanalysis products. We learn that the annual (1yr), interannual (5yrs) and decadal (10 yrs) variability of AMET and OMET anomalies do not agree within the chosen atmospheric and oceanic reanalysis products. For the time scales beyond 10 years, the signals are too short to analyze. It is difficult to provide an answer for the maximum time scale that can be viewed as robust. An easy suggestion is, we have to wait until we have much longer historical records.

*Figures 9 and 10: Having only lines and no stippling, it is sometimes hard to see what areas are significant. For example, in the SIC regressions I cannot tell which areas are significant and which not. I recommend to improve the plots to make this clearer. A standard way would be to stipple significant areas.*

We updated those figures with stippling to indicate the significant areas and adjust the stippling color to avoid masking the shades.

**Figures: 9 and 10 and Figures in the supplementary material**

Again, we'd like to express our gratitude to the time and effort that the reviewer spent on our manuscript.

With best regards

[revised manuscript text omitted]